# Reinforcement Learning with Bayesian Classifiers: Efficient Skill Learning from Outcome Examples

## Abstract

Exploration in reinforcement learning is, in general, a challenging problem. In this work, we study a more tractable class of reinforcement learning problems defined by data that provides examples of successful outcome states. In this case, the reward function can be obtained automatically by training a classifier to classify states as successful or not. We argue that, with appropriate representation and regularization, such a classifier can guide a reinforcement learning algorithm to an effective solution. However, as we will show, this requires the classifier to make uncertainty-aware predictions that are very difficult with standard deep networks. To address this, we propose a novel mechanism for obtaining calibrated uncertainty based on an amortized technique for computing the normalized maximum likelihood distribution. We show that the resulting algorithm has a number of intriguing connections to both count-based exploration methods and prior algorithms for learning reward functions from data, while also being able to guide algorithms towards the specified goal more effectively. We show how using amortized normalized maximum likelihood for reward inference is able to provide effective reward guidance for solving a number of challenging navigation and robotic manipulation tasks which prove difficult for other algorithms.

## 1 Introduction

While reinforcement learning (RL) has been shown to successfully solve problems with careful reward design (Rajeswaran et al., 2018; OpenAI et al., 2019), RL in its most general form, with no assumptions on the dynamics or reward function, requires solving a challenging uninformed search problem in which rewards are sparsely observed. Techniques which explicitly provide "reward-shaping" (Ng et al., 1999), or modify the reward function to guide learning, can help take some of the burden off of exploration, but shaped rewards can be difficult to obtain without domain knowledge.

In this paper, we aim to reformulate the reinforcement learning problem to make it easier for the user to specify the task and to provide a tractable reinforcement learning objective. Instead of requiring a reward function designed for an objective, our method instead assumes a user-provided set of *successful outcome examples*: states in which the desired task has been accomplished successfully. The algorithm aims to estimate the distribution over these states and maximize the probability of reaching states that are likely under the distribution. Prior work on learning from success examples (Fu et al., 2018b; Zhu et al., 2020) focused primarily on alleviating the need for manual reward design. In our work, we focus on the potential for this mode of task specification to produce more tractable RL problems and solve more challenging classes of tasks. Intuitively, when provided with explicit examples of successful states, the RL algorithm should be able to direct its exploration, rather than simply hope to randomly chance upon high reward states.

The main challenge in instantiating this idea into a practical algorithm is performing appropriate uncertainty quantification in estimating whether a given state corresponds to a successful outcome. Our approach trains a classifier to distinguish successful states, provided by the user, from those generated by the current policy, analogously to generative adversarial networks (Goodfellow et al., 2014) and previously proposed methods for inverse reinforcement learning (Fu et al., 2018a). In general, such a classifier is not guaranteed to provide a good optimization landscape for learning

the policy. We discuss how a particular form of uncertainty quantification based on the normalized maximum likelihood (NML) distribution produces better reward guidance for learning. We also connect our approach to count-based exploration methods, showing that a classifier with suitable uncertainty estimates reduces to a count-based exploration method in the absence of any generalization across states, while also discussing how it improves over count-based exploration in the presence of good generalization. We then propose a practical algorithm to train success classifiers in a computationally efficient way with NML, and show how this form of reward inference allows us to solve difficult problems more efficiently, providing experimental results which outperform existing algorithms on a number of navigation and robotic manipulation domains.

## 2 RELATED WORK

A number of techniques have been proposed to improve exploration.These techniques either add reward bonuses that encourage a policy to visit novel states in a task-agnostic manner (Wiering and Schmidhuber, 1998; Auer et al., 2002; Schaul et al., 2011; Houthooft et al., 2016; Pathak et al., 2017; Tang et al., 2017; Stadie et al., 2015; Bellemare et al., 2016; Burda et al., 2018a; O'Donoghue, 2018) or perform Thompson sampling or approximate Thompson sampling based on a prior over value functions (Strens, 2000; Osband et al., 2013; 2016). While these techniques are uninformed about the actual task, we consider a constrained set of problems where examples of successes can allow for more task-directed exploration. In real world problems, designing well-shaped reward functions makes exploration easier but often requires significant domain knowledge (Andrychowicz et al., 2020), access to privileged information about the environment (Levine et al., 2016) and/or a human in the loop providing rewards (Knox and Stone, 2009; Singh et al., 2019b). Prior work has considered specifying rewards by providing example demonstrations and inferring rewards with inverse RL (Abbeel and Ng, 2004; Ziebart et al., 2008; Ho and Ermon, 2016; Fu et al., 2018a). This requires expensive expert demonstrations to be provided to the agent. In contrast, our work has the minimal requirement of simply providing successful outcome states, which can be done cheaply and more intuitively. This subclass of problems is also related to goal conditioned RL (Kaelbling, 1993; Schaul et al., 2015; Zhu et al., 2017; Andrychowicz et al., 2017; Nair et al., 2018; Veeriah et al., 2018; Rauber et al., 2018; Warde-Farley et al., 2018; Colas et al., 2019; Ghosh et al., 2019; Pong et al., 2020) but is more general, since it allows for a more abstract notion of task success.

A core idea behind our work is using a Bayesian classifier to learn a suitable reward function. Bayesian inference with expressive models and high dimensional data can often be intractable, requiring assumptions on the form of the posterior (Hoffman et al., 2013; Blundell et al., 2015; Maddox et al., 2019). In this work, we build on the concept of normalized maximum likelihood (Rissanen, 1996; Shtar'kov, 1987), or NML, to learn Bayesian classifiers. Although NML is typically considered from the perspective of optimal coding (Grünwald, 2007; Fogel and Feder, 2018), we show how it can be used to learn success classifiers, and discuss its connections to exploration and reward shaping in RL.

## 3 PRELIMINARIES

In this paper, we study a modified reinforcement learning problem, where instead of the standard reward function, the agent is provided with *successful outcome examples*. This reformulation not only provides a modality for task specification that may be more natural for users to provide in some settings (Fu et al., 2018b; Zhu et al., 2020; Singh et al., 2019a), but, as we will show, can also make learning easier. We also derive a meta-learned variant of the conditional normalized maximum likelihood (CNML) distribution for representing our reward function, in order to make evaluation tractable. We discuss background on successful outcome examples and CNML in this section.

### 3.1 REINFORCEMENT LEARNING WITH EXAMPLES OF SUCCESSFUL OUTCOMES

We follow the framework proposed by Fu et al. (2018b) and assume that we are provided with a Markov decision process (MDP) *without* a reward function, given by $\mathcal{M}$, where $\mathcal{M} = (\mathcal{S}, \mathcal{A}, \mathcal{T}, \gamma, \mu_0)$, as well as successful outcome examples $\mathcal{S}_+ = \{s_+^k\}_{k=1}^K$, which is a set of states in which the desired task has been accomplished. This formalism is easiest to describe in terms of the control as inference framework (Levine, 2018). The relevant graphical model in Figure 9 consists of states and actions, as well as binary *success* variables $e_t$ which represent the occurrence of a particular

event. The agent's objective is to cause this event to occur (e.g., a robot that is cleaning the floor must cause the "floor is clean" event to occur). Formally, we assume that the states in $\mathcal{S}_+$ are sampled from the distribution $p(s_t|e_t = \text{True})$ – that is, states where the desired event has taken place. In this work, we focus on efficient methods for solving this reformulation of the RL problem, by utilizing a novel uncertainty quantification method to represent the distribution $p(e_t|s_t)$.

In practice, prior methods that build on this and similar reformulations of the RL problem (Fu et al., 2018b) derive an algorithm where the reward function in RL is produced by a classifier that estimates $p(e_t = \text{True}|s_t)$. Following the adversarial inverse reinforcement learning (AIRL) derivation (Fu et al., 2018a; Finn et al., 2016), it is possible to show that the correct source of *negative* examples for training this classifier is the state distribution of the policy itself, $\pi(s)$. This insight results in a simple algorithm: at each iteration of the algorithm, the policy is updated to maximize the current reward, given by $\log p(e_t = \text{True}|s_t)$, then samples from the policy are added to the set of negative examples $\mathcal{S}_-$, and the classifier is retrained on the original positive set $\mathcal{S}_+$ and the updated negative set $\mathcal{S}_-$.

### 3.2 CONDITIONAL NORMALIZED MAXIMUM LIKELIHOOD

Our method builds on the principle of conditional normalized maximum likelihood (NML) (Rissanen and Roos, 2007; Grünwald, 2007; Fogel and Feder, 2018), which we review briefly. CNML is a method for performing $k$-way classification, given a model class $\Theta$ and a dataset $\mathcal{D} = \{(x_0, y_0), (x_1, y_1), ..., (x_n, y_n)\}$, and has been shown to provide better calibrated predictions and uncertainty estimates with minimax regret guarantees (Bibas et al., 2019). To predict the class of a query point $x_q$, CNML constructs $k$ augmented datasets by adding $x_q$ with a different label in each datasets, which we write as $\mathcal{D} \cup (x_q, y = i), i \in (1, 2, ..., k)$. CNML then defines the class distribution by solving the maximum likelihood estimation problem at query time for each of these augmented datasets to convergence, and normalize the likelihoods as follows:

$$p_{\text{CNML}}(y = i|x_q) = \frac{p_{\theta_i}(y = i|x_q)}{\sum_{j=1}^{k} p_{\theta_j}(y = j|x_q)}, \qquad \theta_i = \arg\max_{\theta \in \Theta} \mathbb{E}_{(x,y) \sim \mathcal{D} \cup (x_q, y=i)}[\log p_\theta(y|x)] \quad (1)$$

Intuitively, if $x_q$ is close to other datapoints in $\mathcal{D}$, then the model will struggle to assign a high likelihood to labels that differ substantially from other nearby points. However, if $x_q$ is far from all datapoints in $\mathcal{D}$, then the different augmented MLE problems can easily classify $x_q$ as an arbitrary class, providing us with a likelihood closer to uniform. We refer readers to Grünwald (2007) for an in-depth discussion. A major limitation of CNML is that it requires training an entire neural network to convergence on the entire augmented dataset every time we want to evaluate a test point's class probabilities. We will address this issue in Section 5.

## 4 BAYESIAN SUCCESS CLASSIFIERS FOR REWARD INFERENCE

Ideally, training a classifier with the policy samples as negative examples as described in Section 3.1 should yield a smooth decision boundary between the well-separated negative and positive examples. For example, Figure 2 depicts a simple 1-D scenario, where the agent starts at the left ($s_0$) and the positive outcomes are at the right ($s_+$) side of the environment. Since the positives are on the right and the negatives are on the left, one might expect a classifier to gradually increase its prediction of a success as we move to the right (Figure 2a), which would provide a dense reward signal for the policy to move to the right. However, this idealized scenario rarely happens in practice. With-

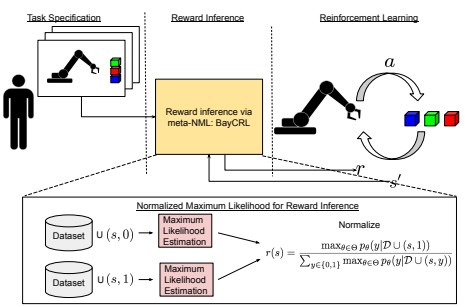

Figure 1: BayCRL: Illustration of how Bayesian classifiers learned with normalized maximum likelihood can be used to provide informative learning signal during learning. The human user provides examples of successful outcomes, which are then used in combination with on policy samples to iterate between training a classifier with NML and training RL with this reward.

out suitable regularization, the decision boundary between the positive and negative examples may

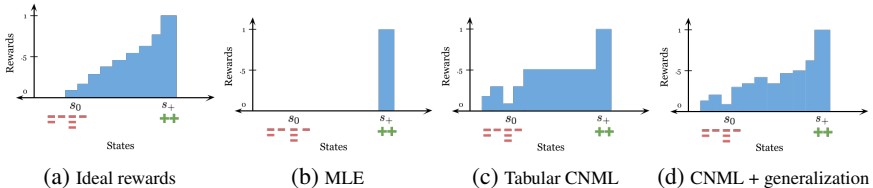

(a) Ideal rewards     (b) MLE     (c) Tabular CNML     (d) CNML + generalization

Figure 2: An idealized illustration of a well-shaped reward and the solutions that various classifier training schemes might provide. Red bars represent visited states near the initial state $s_0$ and green pluses represent the example success examples $s_+$. (a) The ideal reward would provide learning signal to encourage the policy to move from the start states $s_0$ to the successful states $s_+$. (b) When training a success classifier with MLE, the classifier may output zero (or arbitrary) probabilities when evaluated at new states. (c) Tabular CNML will give a prior probability of $0.5$ on new states. (d) When using function approximation, CNML with generalization will provide a degree of shaping towards the goal.

not be smooth. In fact, the decision boundary of an optimal classifier may take on the form of a sharp boundary *anywhere* between the positive and negative examples in the early stages of training (Figure 2b). As a result, the classifier might provide little to no reward signal for the policy, since it can assign arbitrarily small probabilities to the states sampled from the policy. We note that this issue is not pathological: our experiments in Section 6 show that this poor reward signal issue happens in practice and can greatly hinder learning. In this section, we will discuss how an appropriate classifier training method can avoid these uninformative rewards.

## 4.1 REGULARIZED SUCCESS CLASSIFIERS VIA NORMALIZED MAXIMUM LIKELIHOOD

To create effective shaping, we would like our classifier to provide a more informative reward when evaluated at rarely visited states that lie on the path to successful outcomes. A more informative reward function is one that assigns higher rewards to the *fringe* of the states visited by the policy, because this will encourage the policy to explore and move towards the desired states. We can construct such a reward function by imposing the prior

---

**Algorithm 1** RL with CNML-Based Success Classifiers

1: User provides success examples $\mathcal{S}_+$
2: Initialize policy $\pi$, replay buffer $\mathcal{S}_-$, and reward classifier parameters $\theta_\mathcal{R}$
3: **for** iteration $i = 1, 2, ...$ **do**
4:      Add on-policy examples to $\mathcal{S}_-$ by executing $\pi$.
5:      Sample $n_{\text{test}}$ points from $\mathcal{S}_+$ (label 1) and $n_{\text{test}}$ points from $\mathcal{S}_-$ (label 0) to construct a dataset $\mathcal{D}$
6:      Assign state rewards as $r(s) = p_{\text{CNML}}(e = 1|s, \mathcal{D})$
7:      Train $\pi$ with RL algorithm

---

that novel states have a non-negligible chance of being a success state. To do so, we train a Bayesian classifier using conditional normalized maximum likelihood (CNML) (Shtar'kov, 1987), as we described in Section 3, which corresponds to imposing a uniform prior on the output class probabilities.

To use CNML for reward inference, the procedure is similar to the one described in Section 3. We construct a dataset using the provided successful outcomes as positives and the on-policy samples as negatives. However, the label probabilities for RL are then produced by the CNML procedure described in Equation 1 to obtain the rewards $r(s) = p_{\text{CNML}}(e = 1|s)$. To illustrate how this affects reward assignment during learning, we visualize a potential assignment of rewards with a CNML-based classifier on the problem described earlier. When the success classifier is trained with CNML instead of standard maximum likelihood, intermediate unseen states would receive non-zero rewards rather than simply having vanishing likelihoods like in Figure 2b. The didactic illustrations in Fig 2c and Fig 2d show how the rewards obtained via NML might incentivize exploration. In fact, the CNML likelihood corresponds to a form of count-based exploration (as we show below), while also providing more directed shaping towards the goal when generalization exists across states.

## 4.2 RELATIONSHIP TO COUNT-BASED EXPLORATION

In this section we relate the success likelihoods obtained via CNML to commonly used exploration methods based on counts. Formally, we prove that the success classifier trained with CNML is equivalent to a version of count-based exploration with a sparse reward function in the absence of any generalization across states (i.e., a fully tabular setting).

**Theorem 4.1.** *Suppose we are estimating success probabilities $p(e = 1|s)$ in the tabular setting, where we have an independent parameter for each state. Let $N(s)$ denote the number of times state $s$ has been visited by the policy, and let $G(s)$ be the number of occurrences of state $s$ in the set of goal examples. Then the CNML success probability $p_{CNML}(e = 1|s)$ is equal to $\frac{G(s)+1}{N(s)+G(s)+2}$. For states that are not represented in the goal examples, i.e. G(s) = 0, we then recover inverse counts $\frac{1}{N(s)+2}$.*

Refer to Appendix A.7 for a full proof.

### 4.3 REWARD SHAPING WITH BAYESIAN SUCCESS CLASSIFIERS

While the analysis above suggests that a CNML classifier would give us something akin to a sparse reward plus an exploration bonus, the structure of the problem and the state space actually provides us more information to guide us towards the goal. In most environments (Brockman et al., 2016; Yu et al., 2019) the state space does not consist of independent and uncorrelated categorical variables, and is instead provided in a representation that relates at least roughly to the dynamics structure in the environment. For instance, states close to the goal dynamically are also typically close to the goal in the metric space defined by the states. Indeed, this observation is the basis of many commonly used heuristic reward shaping methods, such as rewards given by Euclidean distance to target states.

In this case, the task specification can actually provide more information than simply performing uninformed count-based exploration. Since the uncertainty-aware classifier described in Section 4.1 is built on top of features that are correlated with environment dynamics, and is trained with knowledge of the desired outcomes, it is able to incentivize *task-aware* directed exploration. As compared to CNML without generalization in Fig 2c, we expect the intermediate rewards to provide more shaping towards the goal. This phenomenon is illustrated intuitively in Fig 2d, and visualized and demonstrated empirically in our experimental analysis in Section 6, where BayCRL is able to significantly outperform methods for task-agnostic exploration.

### 4.4 OVERVIEW

In this section, we introduced the idea of Bayesian classifiers trained via CNML as a means to provide rewards for RL problems specified by examples of successful outcomes. Concretely, a CNML-based scheme has the following advantages:

- **Natural exploration behavior due to accurate uncertainty estimation** in the output success probabilities. This is explained by the connection between CNML and count-based exploration in the discrete case, and benefits from additional generalization in practical environments, as we will see in Section 6.
- **Better reward shaping by utilizing goal examples** to guide the agent more quickly and accurately towards the goal. We have established this benefit intuitively, and will validate it empirically through extensive visualizations and experiments in Section 6.

## 5 BAYCRL: TRAINING BAYESIAN SUCCESS CLASSIFIERS FOR OUTCOME DRIVEN RL VIA META-LEARNING AND CNML

In Section 4, we discussed how Bayesian success classifiers can incentivize exploration and provide reward shaping to guide RL. However, the reward inference technique via CNML described in Section 4.1 is computationally intractable, as it requires optimizing maximum likelihood estimation problems to convergence on every data point we want to query. In this section, we describe a novel approximation that allows us to instantiate this method in practice.

### 5.1 META-LEARNING FOR CNML

We adopt ideas from meta-learning to amortize the cost of obtaining the CNML distribution. As noted in Section 4.1, the computation of the CNML distribution involves repeatedly solving maximum likelihood problems. While computationally daunting, these problems share a significant amount of common structure, which we can exploit to quickly obtain CNML estimates. One set of techniques that

are directly applicable is meta-learning for few shot classification. Meta-learning uses a distribution of training problems to explicitly learn models that can quickly adapt to new problems.

To apply meta-learning to the CNML problem, we can formulate each of the maximum likelihood problems described in Equation 1 as a separate task for meta-learning, and apply any standard meta-learning technique to obtain a model capable of few-shot adaptation to the MLE problems required for CNML. While any meta-learning algorithm is applicable, we found model agnostic meta-learning (MAML)(Finn et al. (2017)) to be an effective choice of algorithm. In short, MAML tries to learn a model that can quickly adapt to new tasks via a few steps of gradient descent.

This procedure is illustrated in Fig 10, and can be described as follows: given a dataset $\mathcal{D} = \{(x_0, y_0), (x_1, y_1), ..., (x_n, y_n)\}$, $2n$ different tasks $\tau_i$ can be constructed, each corresponding to performing maximum likelihood estimation on the dataset with a certain proposed label for $x_i$: $\max_\theta \mathbb{E}_{(x,y) \sim \mathcal{D} \cup (x_i, y=0)}[\log p(y|x, \theta)]$ or $\max_\theta \mathbb{E}_{(x,y) \sim \mathcal{D} \cup (x_i, y=1)}[\log p(y|x, \theta)]$. Given these constructed tasks $\mathcal{S}(\tau)$, meta-training as described in Finn et al. (2017):

$$\max_\theta \ \mathbb{E}_{\tau \sim \mathcal{S}(\tau)}[\mathcal{L}(\tau, \theta')], \quad s.t \ \ \theta' = \theta - \alpha \nabla_\theta \mathcal{L}(\tau, \theta). \tag{2}$$

This training procedure gives us parameters $\theta$ that can then be quickly adapted to provide the CNML distribution simply by performing a step of gradient descent. The model can be queried for the CNML distribution by starting from $\theta$ and taking one step of gradient descent for the query point augmented dataset, each with a different potential label. These likelihoods are then normalized to provide the CNML distribution as follows:

$$p_{\text{meta-NML}}(y|x; \mathcal{D}) = \frac{p_{\theta_y}(y|x)}{\sum_{y \in \mathcal{Y}} p_{\theta_y}(y|x)}, \quad \theta_y = \theta - \alpha \nabla_\theta \mathbb{E}_{(x_i, y_i) \sim \mathcal{D} \cup (x, y)}[\mathcal{L}(x_i, y_i, \theta)]. \tag{3}$$

This algorithm, which we call *meta-NML*, allows us to obtain normalized likelihood estimates without having to retrain maximum likelihood to convergence at every single query point, since the model can now solve maximum likelihood problems of this form very quickly. A complete detailed description and pseudocode of this algorithm are provided in Appendix A.2.

Crucially, we find that meta-NML is able to approximate the true NML outputs with just one or a few gradient steps.

|  | **Feedforward** | **Meta-NML** | **Naive CNML** |
|---|---|---|---|
| **Single input point** | 0.0004s | 0.0090s | 15.19s |
| **Epoch of RL** | 23.50s | 39.05s | 4hr 13min 34s |

Table 1: Runtimes for evaluating a single input point and running one epoch of RL using feedforward, meta-NML, and naive CNML classifiers. Meta-NML provides over a 1600x speedup compared to naive CNML. For a full list of runtimes see Appendix A.3.3.

This makes it several orders of magnitude faster than naive CNML, which would normally require multiple passes through the entire dataset on each input point in order to train to convergence.

## 5.2 Applying Meta-NML to Success Classification

We apply the meta-NML algorithm described above to learning Bayesian success classifiers for providing rewards for reinforcement learning, in our proposed algorithm, which we term BayCRL— Bayesian classifiers for reinforcement learning. Similarly to Fu et al. (2018b), we can train our Bayesian classifier by first constructing a dataset $\mathcal{D}$ for binary classification. This is done by using the provided examples of successful outcomes as positives, and on-policy examples collected by the policy as negatives, balancing the number of sam-

---

**Algorithm 2** BayCRL: Bayesian Classifiers for RL

1: User provides success examples $\mathcal{S}_+$
2: Initialize policy $\pi$, replay buffer $\mathcal{S}_-$, and reward classifier parameters $\theta_{\mathcal{R}}$
3: **for** iteration $i = 1, 2, ...$ **do**
4:     Collect on-policy examples to add to $\mathcal{S}_-$ by executing $\pi$.
5:     **if** iteration $i \mod k == 0$ **then**
6:         Sample $n_{\text{train}}$ states from $\mathcal{S}_-$ to create $2n_{\text{train}}$ meta-training tasks
7:         Sample $n_{\text{test}}$ total test points equally from $\mathcal{S}_+$ (label 1) and $\mathcal{S}_-$ (label 0)
8:         Meta-train $\theta_{\mathcal{R}}$ via meta-NML using Equation 2
9:     Assign state rewards via Equation 4
10:     Train $\pi$ with RL algorithm

---

pled positives and negatives in the dataset. Given this dataset, the Bayesian classifier parameters $\theta_{\mathcal{R}}$ can be trained via meta-NML as described in Equation 2. The classifier can then be used to directly and quickly assign rewards to a state $s$ according to its probabilities $r(s) = p_{\text{meta-NML}}(e = 1|s)$ (via a step of gradient descent, as described in Equation 4), and perform standard reinforcement learning.

$$p_{\text{meta-NML}}(e = 1|s) = \frac{p_{\theta_1}(e = 1|s)}{\sum_{i \in \{0,1\}} p_{\theta_i}(e = i|s)} \tag{4}$$

$$\theta_i = \theta_{\mathcal{R}} - \alpha \nabla_\theta \mathbb{E}_{(s_j, e_j) \sim \mathcal{D} \cup (s, e=i)}[\mathcal{L}(e_j, s_j, \theta)], \text{ for } i \in \{0,1\} \tag{5}$$

An overview of this algorithm is provided in Algorithm 2, and full details are in Appendix A.2. The rewards start off at an uninformative value of $0.5$ for all unvisited states at the beginning, and close to 1 for successful outcomes. As training progresses, more states are visited, added to the buffer and BayCRL starts to assign them progressively lower reward as they get visited more and more, thereby encouraging visiting of under-visited states. At convergence, all the non successful states will have a reward of close to 0 and states at the goal will have a reward of $0.5$, since the numbers of positive and negative labels for successful outcomes will be balanced as described above.

## 6 EXPERIMENTAL EVALUATION

In our experimental evaluation we aim to answer the following questions: (1) Do the learning dynamics of prior classifier-based reward learning methods provide informative rewards for RL? (2) Does using BayCRL help address the exploration challenge when solving RL problems specified by successful outcomes? (3) Does using BayCRL help provide better reward shaping than simply performing naïvely uninformed exploration? To

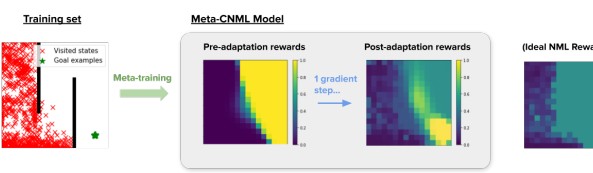

Figure 3: Diagram of using meta-NML to train a classifier. Meta-NML learns an initialization that can quickly adapt to new datapoints with arbitrary labels. At evaluation time, it approximates the NML probabilities (right) fairly well with a single gradient step

evaluate these questions, we evaluate our proposed algorithm BayCRL with the following setup. Further details and videos can be found at https://sites.google.com/view/baycrl/home

### 6.1 EXPERIMENTAL SETUP

We start off by understanding the algorithm behavior by evaluating it on maze navigation problems, which require avoiding several local optima before truly reaching the goal. Then, to evaluate our method in more complex domains, we consider three robotic manipulation tasks that were previously covered in Singh et al. (2019a) with a Sawyer robot arm: door opening, tabletop object pushing, and 3D object picking. As we show in our results, exploration in these environments is challenging and using naively chosen reward shaping often does not solve the problem at hand. More details on each environment and their associated challenges are available in Appendix A.4.1.

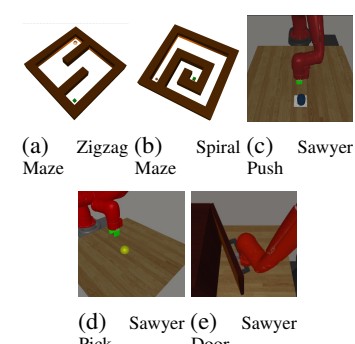

(a) Zigzag Maze (b) Spiral Maze (c) Sawyer Push

(d) Sawyer Pick (e) Sawyer Door

Figure 4: We evaluate on two maze environments and three robotics tasks: (a) the agent must navigate around an S-shaped corridor, (b) the agent must navigate a spiral corridor, (c) the robot must push a puck to location, (d) the robot must raise a randomly placed tennis ball to location, (e) the robot must open the door a specified angle.

We compare with a number of prior algorithms and ablations. To provide a comparison with a standard previous method which uses success classifiers trained with an IRL-based adversarial method, we include the VICE algorithm (Fu et al., 2018b). Note that this algorithm is quite related to BayCRL, but it uses a standard maximum likelihood classifier rather than a Bayesian classifier trained with CNML and meta-learning. We also include a comparison with DDL, a recently proposed technique for learning dynamical distances (Hartikainen et al., 2019). We additionally

include comparisons to algorithms for uninformed exploration to show that BayCRL does a more directed form of exploration and reward shaping. To provide an apples-to-apples comparison, we use the same VICE method for training classifiers, but combine it with novelty-based exploration based on random network distillation (Burda et al., 2018b) for the robotic manipulation tasks, and oracle inverse count bonuses for the maze navigation tasks. Finally, to demonstrate the importance of well-shaped rewards, we compare to running Soft Actor-Critic (Haarnoja et al., 2018), a standard RL algorithm for continuous domains, with two naive reward functions: a sparse reward at the goal, and a heuristically shaped reward which uses L2 distance to the goal state. More details on each algorithm and the hyperparameters used are included in Appendix A.6.

## 6.2 Comparisons with Prior Algorithms

We compare with prior algorithms on the domains described above. As we can see in Fig 5, BayCRL is able to very quickly learn how to solve these challenging exploration tasks, often reaching better asymptotic performance than most prior methods, and doing so more efficiently than VICE (Fu et al., 2018b) or DDL (Hartikainen et al., 2019). This suggests that BayCRL is able to provide directed reward shaping and exploration that is substantially better than standard classifier-based methods (e.g., VICE).

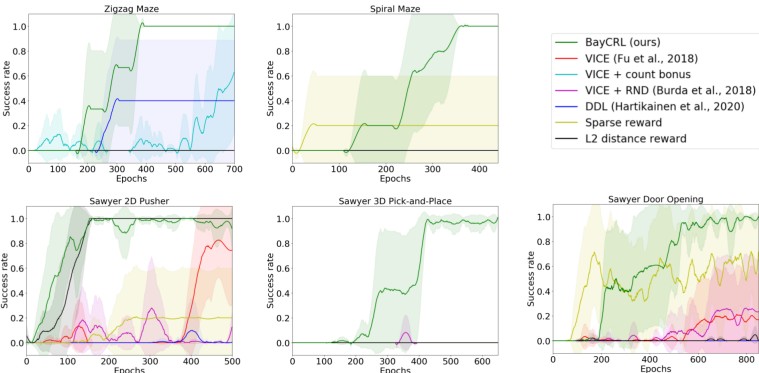

Figure 5: BayCRL outperforms prior goal-reaching methods on all our evaluation environments. BayCRL also performs better or comparably to a heuristically shaped hand-designed reward that uses Euclidean distance, demonstrating that designing a well-shaped reward is not trivial in these domains. Shading indicates a standard deviation across 5 seeds. For details on the success metrics used, see Appendix A.4.2.

To isolate whether the benefits purely come from exploration or also from task-aware reward shaping, we compare with methods that only perform uninformed, task-agnostic exploration. On the maze environments, where we can discretize the state space, we compute ground truth count-based bonuses for exploration. For the higher dimensional robotics tasks, we use RND (Burda et al., 2018b). From these comparisons, shown in Fig 5, it is clear that BayCRL significantly outperforms methods that use novelty-seeking exploration, but do not otherwise provide effective reward shaping. In combination with our visualizations in Section 6.4, this suggests that BayCRL is providing useful task-aware reward shaping more effectively than uniformed exploration methods. We also compare BayCRL to a manually heuristically-designed shaped reward function, based on Euclidean distance. As shown in Fig 5, BayCRL generally outperforms simple manual shaping in terms of sample complexity and asymptotic performance, indicating that the learned shaping is non-trivial and adapted to the task.

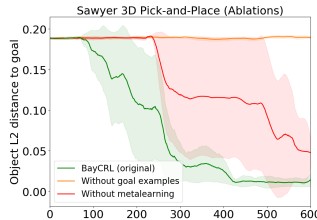

Figure 6: Ablative analysis of BayCRL. The amortization from meta-learning and access to goal examples are both important components for performance.

### 6.3 ABLATIONS

We first evaluate the importance of meta-learning for estimating the NML distribution. In Figure 6, we see that naively estimating the NML distribution by taking a single gradient step and following the same process as evaluating meta-NML, but without any meta-training, results in much worse performance. Second, we analyze the importance of making the BayCRL classifier aware of the task being solved, to understand whether BayCRL is informed by the success examples or simply approximates count-based exploration. To that end, we modify the training procedure so that the dataset $\mathcal{D}$ consists of only the on-policy negatives, and add the inferred reward from the Bayesian classifier to the reward obtained by a standard MLE classifier (similarly to the VICE+RND baseline). We see that this performs poorly, showing that the BayCRL classifier is doing more than just performing count-based exploration, and benefits from better reward shaping due to the provided goal examples. Further ablations are available in Appendix A.5.

### 6.4 ANALYSIS OF BAYCRL

**BayCRL and Reward Shaping.** To better understand how BayCRL provides reward shaping, we visualize the rewards for various slices along the $z$ axis on the Sawyer Pick task, an environment which presents a significant exploration challenge. In Fig 7 we see that the BayCRL rewards clearly correlate with the distance to the object's goal position, shown as a white star, thus guiding the robot to raise the ball to the desired location even if it has never reached the goal before. In contrast, the MLE classifier has a sharp, poorly-shaped decision boundary.

**BayCRL and Exploration.** Next, to illustrate the connection between BayCRL and exploration, we compare the states visited by BayCRL (which uses a meta-NML classifier) and by VICE (which uses a standard L2-regularized classifier) in Figure 8. We see that BayCRL naturally incentivizes the agent to visit novel states, allowing it to navigate around local minima and reach the true goal. In contrast, VICE learns a misleading reward function that prioritizes closeness to the goal in $xy$ space, causing the agent to stay on the wrong side of the wall.

Interestingly, despite incentivizing exploration, BayCRL does not simply visit all possible states; at convergence, it has only covered around 70% of the state space. In fact, we see in the scatterplots in Figure 8 that BayCRL prioritizes states that bring it closer to the goal and ignores ones that don't, thus making use of the goal examples provided to it. This suggests that BayCRL benefits from a *combination* of novelty-seeking behavior and effective reward shaping, allowing it to choose new states strategically.

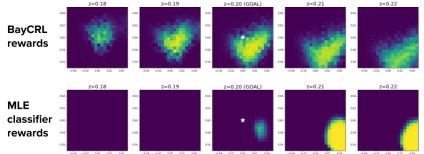

Figure 7: Visualization of reward shaping for 3D Pick-and-Place at various z values (heights). BayCRL learns rewards that provide a smooth slope toward the goal, adapting to the policy and guiding it to learn the task, while the MLE classifier learns a sharp and poorly shaped decision boundary.

## 7 DISCUSSION

In this work, we consider a subclass of reinforcement learning problems where examples of successful outcomes specify the task. We analyze how solutions via standard success classifiers suffer from shortcomings, and training Bayesian classifiers allows for better exploration to solve challenging problems. We discuss how the NML distribution can provide us a way to train such Bayesian classifiers, providing benefits of exploration and reward shaping. To make learning tractable, we propose a novel meta-learning approach to amortize the NML process.

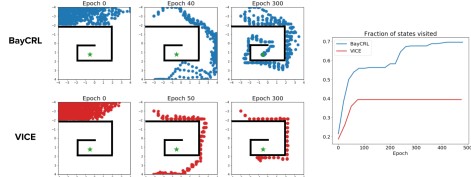

Figure 8: Plots of visitations and state coverage over time for BayCRL vs. VICE. BayCRL explores a significantly larger portion of the state space and is able to avoid local optima.

While this work has shown the effectiveness of Bayesian classifiers for reward inference for tasks in simulation, it would be interesting to scale this solution to real world problems. Additionally, obtaining a theoretical understanding of how reward shaping interacts with learning dynamics would be illuminating in designing reward schemes.

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

## A APPENDIX

### A.1 GRAPHICAL MODEL FOR CONTROL AS INFERENCE

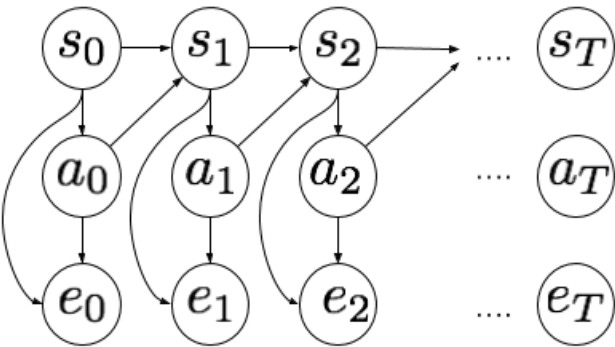

Figure 9: Graphical Model framework for Control as Inference. $e_t$ correspond to auxiliary event variables representing successfully accomplishing the task

### A.2 DETAILED DESCRIPTION OF META-NML

We provide a detailed description of the meta-NML algorithm described in Section 5, and the details of the practical algorithm.

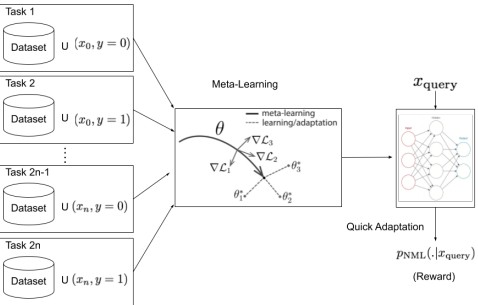

Figure 10: Figure illustrating the meta-training procedure for meta-NML.

Given a dataset $\mathcal{D} = \{(x_0, y_0), (x_1, y_1), .., (x_n, y_n)\}$, the meta-NML procedure proceeds by first constructing $k * n$ tasks from these data points, for a $k$ shot classification problem. We will keep $k = 2$ for simplicity in this description, in accordance with the setup of binary success classifiers in RL. Each task $\tau_i$ is constructed by augmenting the dataset with a negative label $\mathcal{D} \cup (x_i, y = 0)$ or a positive label $\mathcal{D} \cup (x_i, y = 1)$. Now that each task consists of solving the maximum likelihood problem for its augmented dataset, we can directly apply standard meta-learning algorithms to this setting. Building off the ideas in MAML (Finn et al., 2017), we can then train a set of model parameters $\theta$ such that after a single step of gradient descent it can quickly adapt to the optimal solution for the MLE problem on any of the augmented datasets. This is more formally written as

$$\max_{\theta} \ \mathbb{E}_{\tau \sim \mathcal{S}(\tau)}[\mathcal{L}(\tau, \theta')], \quad s.t \ \theta' = \theta - \alpha \nabla_\theta \mathcal{L}(\tau, \theta) \quad (6)$$

where $\mathcal{L}$ represents a standard classification loss function, $\alpha$ is the learning rate, and the distribution of tasks $p(\tau)$ is constructed as described above. For a new query point $x$, these initial parameters can then quickly be adapted to provide the CNML distribution by taking a gradient step on each

augmented dataset to obtain the approximately optimal MLE solution, and normalizing these as follows:

$$p_{\text{meta-NML}}(y|x; \mathcal{D}) = \frac{p_{\theta_y}(y|x)}{\sum_{y \in \mathcal{Y}} p_{\theta_y}(y|x)}, \qquad \theta_y = \theta - \alpha \nabla_\theta \mathbb{E}_{(x_i, y_i) \sim \mathcal{D} \cup (x,y)}[\mathcal{L}(x_i, y_i, \theta)]$$

This algorithm in principle can be optimized using any standard stochastic optimization method such as SGD, as described in Finn et al. (2017), backpropagating through the inner loop gradient update. For the specific problem setting that we consider, we have to employ some optimization tricks in order to enable learning:

### A.2.1 IMPORTANCE WEIGHTING ON QUERY POINT

Since only one datapoint is augmented to the training set at query time for CNML, it can get challenging for stochastic gradient descent to pay attention to this datapoint with increasing dataset sizes. For example, if we train on an augmented dataset of size 2048 by cycling through it in batch sizes of 32, then only 1 in 64 batches would include the query point itself and allow the model to adapt to the proposed label, while the others would lead to noise in the optimization process, potentially worsening the model's prediction on the query point.

In order to make sure the optimization considers the query point, we include the query point and proposed label $(x_q, y)$ in every minibatch that is sampled, but downweight the loss computed on that point such that the overall objective remains unbiased. This is simply doing importance weighting, with the query point downweighted by a factor of $\lceil \frac{b-1}{N} \rceil$ where $b$ is the desired batch size and $N$ is the total number of points in the original dataset.

To see why the optimization objective remains the same, we can consider the overall loss over the dataset. Let $f_\theta$ be our classifier, $\mathcal{L}$ be our loss function, $\mathcal{D}' = \{(x_i, y_i)\}_{i=1}^N \cup (x_q, y)$ be our augmented dataset, and $\mathcal{B}_k$ be the $k$th batch seen during training. Using standard SGD training that cycles through batches in the dataset, the overall loss on the augmented dataset would be:

$$\mathcal{L}(\mathcal{D}') = \left( \sum_{i=0}^N \mathcal{L}(f_\theta(x_i), y_i) \right) + \mathcal{L}(f_\theta(x_q), y)$$

If we instead included the downweighted query point in every batch, the overall loss would be:

$$\mathcal{L}(\mathcal{D}') = \sum_{k=0}^{\lceil \frac{b-1}{N} \rceil} \sum_{(x_i, y_i) \in \mathcal{B}_k} \left( \mathcal{L}(f_\theta(x_i), y_i) + \frac{1}{\lceil \frac{b-1}{N} \rceil} \mathcal{L}(f_\theta(x_q), y) \right)$$

$$= \left( \sum_{k=0}^{\lceil \frac{b-1}{N} \rceil} \sum_{(x_i, y_i) \in \mathcal{B}_k} \mathcal{L}(f_\theta(x_i), y_i) \right) + \lceil \frac{b-1}{N} \rceil \frac{1}{\lceil \frac{b-1}{N} \rceil} \mathcal{L}(f_\theta(x_q), y)$$

$$= \left( \sum_{i=0}^N \mathcal{L}(f_\theta(x_i), y_i) \right) + \mathcal{L}(f_\theta(x_q), y)$$

which is the same objective as before.

This trick has the effect of still optimizing the same max likelihood problem required by CNML, but significantly reducing the variance of the query point predictions as we take additional gradient steps at query time. As a concrete example, consider querying a meta-CNML classifier on the input shown in Figure 11. If we adapt to the augmented dataset without including the query point in every batch (i.e. without importance weighting), we see that the query point loss is significantly more unstable, requiring us to take more gradient steps to converge.

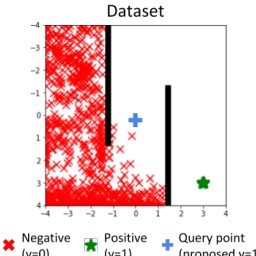 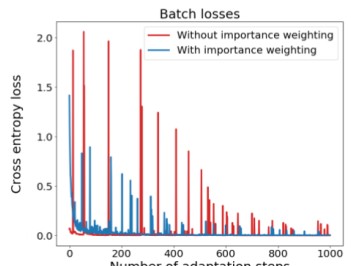 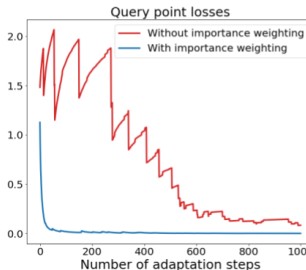

Figure 11: Comparison of adapting to a query point (pictured on left with the original dataset) at test time for CNML with and without importance weighting. The version without importance weighting is more unstable both in terms of overall batch loss and the individual query point loss, and thus takes longer to converge. The spikes in the red lines occur when that particular batch happens to include the query point, since that point's proposed label ($y = 1$) is different than those of nearby points ($y = 0$). The version with importance weighting does not suffer from this problem because it accounts for the query point in each gradient step, while keeping the optimization objective the same.

### A.2.2 KERNEL WEIGHTED TRAINING LOSS

The augmented dataset consists of points from the original dataset $\mathcal{D}$ and one augmented point $(x_q, y)$. Given that we mostly care about having the proper likelihood on the query point, with an imperfect optimization process, the meta-training can yield solutions that are not very accurately representing true likelihoods on the query point. To counter this, we introduce a kernel weighting into the loss function in Equation 6 during meta-training and subsequently meta-testing. The kernel weighting modifies the training loss function as:

$$\max_{\theta} \ \mathbb{E}_{\tau \sim \mathcal{S}(\tau)}[\mathbb{E}_{(x,y)\sim\tau}\mathcal{K}(x, x_\tau)\mathcal{L}(x, y, \theta')], \quad s.t \ \theta' = \theta - \alpha\nabla_\theta\mathbb{E}_{(x,y)\sim\tau}\mathcal{K}(x, x_\tau)\mathcal{L}(x, y, \theta) \quad (7)$$

where $x_\tau$ is the query point for task $\tau$ and $\mathcal{K}$ is a choice of kernel. We typically choose exponential kernels centered around $x_\tau$. Intuitively, this allows the meta-optimization to mainly consider the datapoints that are copies of the query point in the dataset, or are similar to the query point, and ensures that they have the correct likelihoods, instead of receiving interfering gradient signals from the many other points in the dataset. To make hyperparameter selection intuitive, we designate the strength of the exponential kernel by a parameter $\lambda_{dist}$, which is the Euclidean distance away from the query point at which the weight becomes 0.1. Formally, the weight of a point $x$ in the loss function for query point $x_\tau$ is computed as:

$$K(x, x_\tau) = \exp\{-\frac{2.3}{\lambda_{dist}}||x - x_\tau||_2\} \quad (8)$$

### A.2.3 META-TRAINING AT FIXED INTERVALS

While in principle meta-NML would retrain with every new datapoint, in practice we retrain meta-NML once every $k$ epochs. (In all of our experiments we set $k = 1$, but we could optionally increase $k$ if we do not expect the meta-task distribution to change much between epochs.) We warm-start the meta-learner parameters from the previous iteration of meta-learning, so every instance of meta-training only requires a few steps. We find that this periodic training is a reasonable enough approximation, as evidenced by the strong performance of BayCRL in our experimental results in Section 6.

### A.3 META-NML VISUALIZATIONS

#### A.3.1 META-NML WITH ADDITIONAL GRADIENT STEPS

Below, we show a more detailed visualization of meta-NML outputs on data from the Zigzag Maze task, and how these outputs change with additional gradient steps. For comparison, we also include the idealized NML rewards, which come from a discrete count-based classifier.

Meta-NML is able to resemble the ideal NML rewards fairly well with just 1 gradient step, providing both an approximation of a count-based exploration bonus and better shaping towards the goal due to generalization. By taking additional gradient steps, meta-NML can get arbitrarily close to the true NML outputs, which themselves correspond to inverse counts of $\frac{1}{n+2}$ as explained in Theorem 4.1. While this would give us more accurate NML estimates, in practice we found that taking one gradient step was sufficient to achieve good performance on our RL tasks.

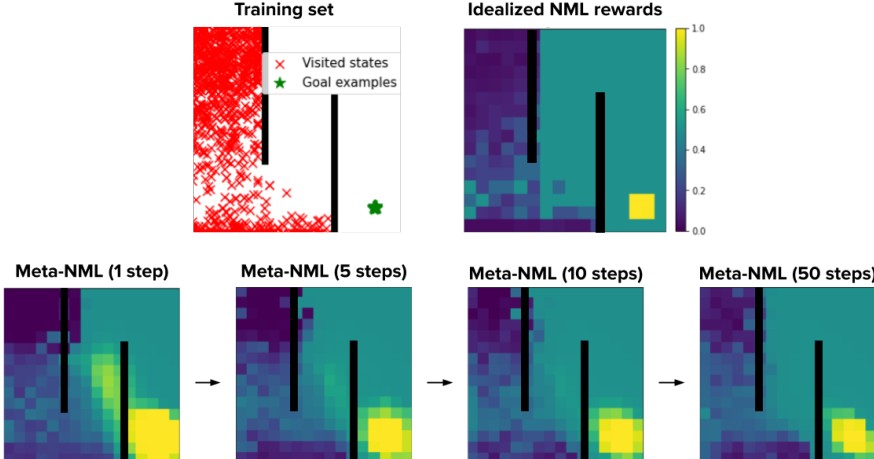

Figure 12: Comparison of idealized (discrete) NML and meta-NML rewards on data from the Zigzag Maze Task. Meta-NML approximates NML reasonably well with just one gradient step at test time, and converges to the true values with additional steps.

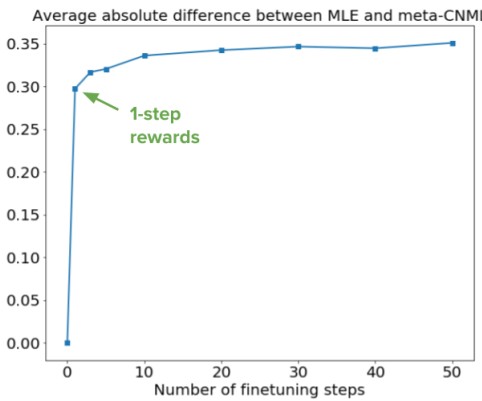

Figure 13: Average absolute difference between MLE and meta-NML goal probabilities across the entire maze state space from Figure 12 above. We see that meta-NML learns a model initialization whose parameters can change significantly in a small number of gradient steps. Additionally, most of this change comes from the first gradient step (indicated by the green arrow), which justifies our choice to use only a single gradient step when evaluating meta-NML probabilities for BayCRL.

### A.3.2 COMPARISON OF REWARD CLASSIFIERS



Figure 14: A comparison of the rewards given by various classifier training schemes on the 2D Zigzag maze. From left to right: (1) An MLE classifier when trained to convergence reduces to an uninformative sparse reward; (2) An MLE classifier trained with regularization and early stopping has smoother contours, but does not accurately identify the goal; (3) The idealized NML rewards correspond to inverse counts, thus providing a natural exploration objective in the absence of generalization; (4) The meta-NML rewards approximate the idealized rewards well in visited regions, while also benefitting from better shaping towards the goal due to generalization.

### A.3.3 RUNTIME COMPARISONS

Below provide the runtimes for feedforward inference, naive CNML, and meta-NML on each of our evaluation domains. We list both the runtimes for evaluating a single input, and for completing a full epoch of training during RL.

These benchmarks were performed on an NVIDIA Titan X Pascal GPU. Per-input runtimes are averaged across 100 samples, and per-epoch runtimes are averaged across 20 epochs.

|  | **Feedforward Inference** | **Meta-NML** | **Naive CNML** |
|---|---|---|---|
| **Mazes (zigzag, spiral)** | 0.0004s | 0.0090s | 15.19s |
| **Sawyer 2D Pusher** | 0.0004s | 0.0092s | 20.64s |
| **Sawyer Door** | 0.0004s | 0.0094s | 20.68s |
| **Sawyer 3D Pick** | 0.0005s | 0.0089s | 20.68s |

Table 2: Runtimes for evaluating a single input point using feedforward, meta-NML, and naive CNML classifiers. Meta-NML provides anywhere between a 1600x and 2300x speedup compared to naive CNML, which is crucial to making our NML-based reward classifier scheme feasible on RL problems.

|  | **Feedforward Inference** | **Meta-NML** | **Naive CNML** |
|---|---|---|---|
| **Mazes (zigzag, spiral)** | 23.50s | 39.05s | 4hr 13min 34s |
| **Sawyer 2D Pusher** | 24.91s | 43.81 | 5hr 44min 25s |
| **Sawyer Door** | 19.77s | 38.52s | 5hr 45min 00s |
| **Sawyer 3D Pick** | 20.24s | 40.73s | 5hr 45min 00s |

Table 3: Runtimes for completing a single epoch of RL according to Algorithm 2. We collect 1000 samples in the environment with the current policy for each epoch of training. The naive CNML runtimes are extrapolated based on the per-input runtime in the previous table. These times indicate that naive CNML would be computationally infeasible to run in an RL algorithm, whereas meta-NML is able to achieve performance much closer to that of an ordinary feedforward classifier and make learning possible.

### A.4 EXPERIMENTAL DETAILS

### A.4.1 ENVIRONMENTS

**Zigzag Maze and Spiral Maze:** These two navigation tasks require moving through long corridors and avoiding several local optima in order to reach the goal. For example, on Spiral Maze, the agent must not get stuck on the other side of the inner wall, even though that position would be close in L2

distance to the desired goal. On these tasks, a sparse reward is not informative enough for learning, while ordinary classifier methods get stuck in local optima due to poor shaping near the goal.

Both of these environments have a continuous state space consisting of the $(x, y)$ coordinates of the agent, ranging from $(-4, -4)$ to $(4, 4)$ inclusive. The action space is the desired velocity in the $x$ and $y$ directions, each ranging from $-1$ to $1$ inclusive.

**Sawyer 2D Pusher:** This task involves using a Sawyer arm, constrained to move only in the $xy$ plane, to push a randomly initialized puck to a fixed location on a table. The state space consists of the $(x, y, z)$ coordinates of the robot end effector and the $(x, y)$ coordinates of the puck. The action space is the desired $x$ and $y$ velocities of the arm.

**Sawyer Door Opening:** In this task, the Sawyer arm is attached to a hook, which it must use to open a door to a desired angle of 45 degrees. The door is randomly initialized each time to be at a starting angle of between 0 and 15 degrees. The state space consists of the $(x, y, z)$ coordinates of the end effector and the door angle (in radians); the action space consists of $(x, y, z)$ velocities.

**Sawyer 3D Pick and Place:** The Sawyer robot must pick up a ball, which is randomly placed somewhere on the table each time, and raise it to a fixed $(x, y, z)$ location high above the table. This represents the biggest exploration challenge out of all the manipulation tasks, as the state space is large and the agent would normally not receive any learning signal unless it happened to pick up the ball and raise it, which is unlikely without careful reward shaping.

The state space consists of the $(x, y, z)$ coordinates of the end effector, the $(x, y, z)$ coordinates of the ball, and the tightness of the gripper (a continuous value between 0 and 1). The robot can control its $(x, y, z)$ arm velocity as well as the gripper value.

### A.4.2   GROUND TRUTH DISTANCE METRICS

In addition to the success rate plots in Figure 5, we provide plots of each algorithm's distance to the goal over time according to environment-specific distance metrics. The distance metrics and success thresholds, which were used to compute the success rates in Figure 5, are listed in the table below.

| Environment | Distance Metric Used | Success Threshold |
|---|---|---|
| Zigzag Maze | Manhattan distance to goal | 0.5 |
| Spiral Maze | Manhattan distance to goal | 0.5 |
| Sawyer 2D Pusher | Puck L2 distance to goal | 0.05 |
| Sawyer Door Opening | Angle difference to goal (radians) | 0.035 |
| Sawyer 3D Pick-and-Place | Ball L2 distance to goal | 0.06 |

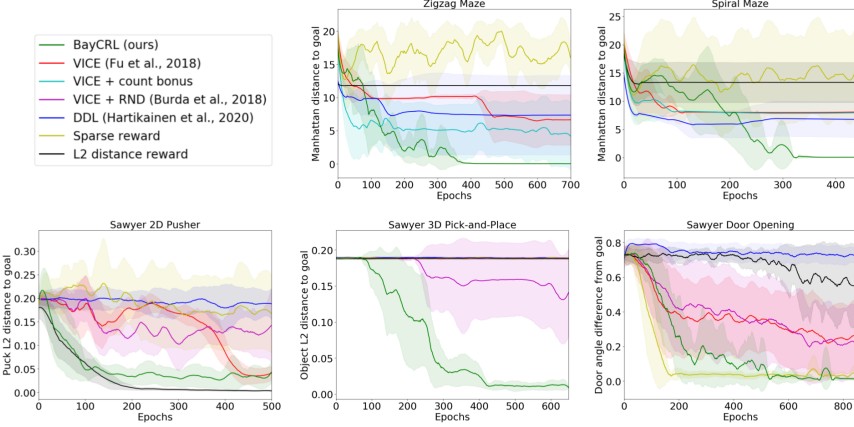

Figure 15: Performance of BayCRL compared to other algorithms according to ground truth distance metrics. We note that while other algorithms seem to be making progress according to these distances, they are often actually getting stuck in local minima, as indicated by the success rates in Figure 5 and the visitation plots in Figure 8.

### A.5 Additional Ablations

#### A.5.1 Learning in a Discrete, Randomized Environment

In practice, many continuous RL environments such as the ones we consider in Section 6 have state spaces that are correlated at least roughly with the dynamics. For instance, states that are closer together dynamically are also typically closer in the metric space defined by the states. This correlation does not need to be perfect, but as long as it exists, BayCRL can in principle learn a smoothly shaped reward towards the goal.

However, even in the case where states are unstructured and completely lack identity, such as in a discrete gridworld environment, the CNML classifier would still reduce to providing an exploration-centric reward bonus, as indicated by Theorem 4.1, ensuring reasonable worst-case performance.

To demonstrate this, we evaluate BayCRL on a variant of the Zigzag Maze task where states are first discretized to a $16 \times 16$ grid, then "shuffled" so that the $xy$ representation of a state does not correspond to its true coordinates and the states are not correlated dynamically. BayCRL manages to solve the task, while a standard classifier method (VICE) does not. Still, BayCRL is more effective in the original state space where generalization is possible, suggesting that both the exploration and reward shaping abilities of the CNML classifier are crucial to its overall performance.

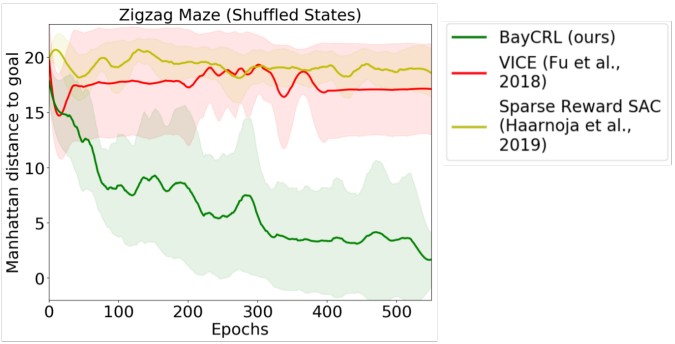

Figure 16: Comparison of BayCRL, VICE, and SAC with sparse rewards on a discrete, randomized variant of the Zigzag Maze task. BayCRL is still able to solve the task on a majority of runs due to its connection to a count-based exploration bonus, whereas ordinary classifier methods (i.e. VICE) experience significantly degraded performance in the absence of any generalization across states.

#### A.5.2 Finding "Hidden" Rewards Not Indicated by Success Examples

The intended setup for BayCRL (and classifier-based RL algorithms in general) is to provide a set of success examples to learn from, thus removing the need for a manually specified reward function. However, here we instead consider the case where a ground truth reward function exists which we do not fully know, and can only query through interaction with the environment. In this case, because the human expert has limited knowledge, the provided success examples may not cover all regions of the state space with high reward.

An additional advantage of BayCRL is that it is still capable of finding these "unspecified" goals because of its built-in exploration behavior, whereas other classifier methods would operate solely based on the goal examples provided. To see this, we evaluate our algorithm on a two-sided variant of the Zigzag Maze with multiple goals, visualized in Figure 17 to the right. The agent starts in the middle and is provided with 5 goal examples on the far left side of the maze; unknown to it, the right side contains 5 sparse reward regions which are actually closer from its initial position.

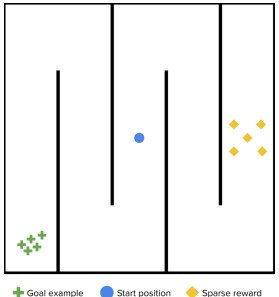

Figure 17: Visualization of the Double-Sided Maze environment. Only the goal examples in the bottom left corner are provided to the algorithm.

As shown in Figures 18 and 19, BayCRL manages to find the sparse rewards while other methods do not. BayCRL, although initially guided towards the provided goal examples on the left, continues to explore in both directions and eventually finds the "hidden" rewards on the right. Meanwhile, VICE focuses solely on the provided goals, and gets stuck in a local optima near the bottom left corner.

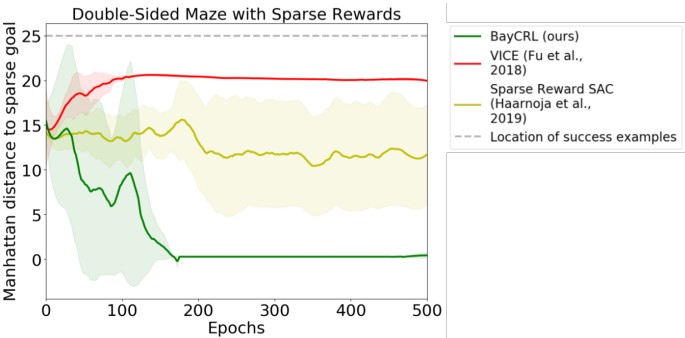

Figure 18: Performance of BayCRL, VICE, and SAC with sparse rewards on a double-sided maze where some sparse reward states are not provided as goal examples. BayCRL is still able to find the sparse rewards, thus receiving higher overall reward, whereas ordinary classifier methods (i.e. VICE) move only towards the provided examples and thus are never able to find the additional rewards. Standard SAC with sparse rewards, also included for comparison, is generally unable to find the goals. The dashed gray line represents the location of the goal examples initially provided to both BayCRL and VICE.

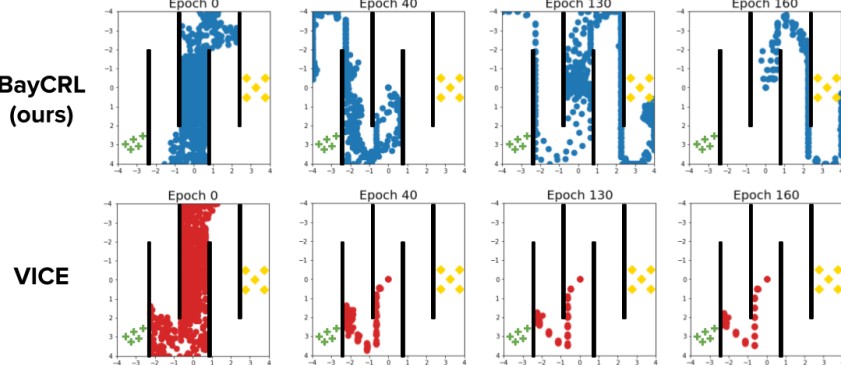

Figure 19: Plot of visitations for BayCRL vs. VICE on the double-sided maze task. BayCRL is initially guided towards the provided goals in the bottom left corner as expected, but continues to explore in both directions, thus allowing it to find the hidden sparse rewards as well. Once this happens, it focuses on the right side of the maze instead because those rewards are easier to reach. In contrast, VICE moves only towards the (incomplete) set of provided goals on the left, ignoring the right half of the maze entirely and quickly getting stuck in a local optima.

A.6   HYPERPARAMETER AND IMPLEMENTATION DETAILS

We describe the hyperparameter choices and implementation details for our experiments here. We first list the general hyperparameters that were shared across runs, then provide tables of additional hyperparameters we tuned over for each domain and algorithm.

| **SAC** | |
|---|---|
| Learning Rate | $3 \times 10^{-4}$ |
| Discount Factor $\gamma$ | 0.99 |
| Policy Type | Gaussian |
| Policy Hidden Sizes | $(512, 512)$ |
| Policy Hidden Activation | ReLU |
| RL Batch Size | 1024 |
| Reward Scaling | 1 |
| Replay Buffer Size | $500,000$ |
| Q Hidden Sizes | $(512, 512)$ |
| Q Hidden Activation | ReLU |
| Q Weight Decay | 0 |
| Q Learning Rate | $3 \times 10^{-4}$ |
| Target Network $\tau$ | $5 \times 10^{-3}$ |
| **BayCRL** | |
| Adaptation batch size | 64 |
| Meta-training tasks per epoch | 128 |
| Meta-test set size | 2048 |
| **VICE** | |
| Classifier Learning Rate | $1 \times 10^{-4}$ |
| Classifier Batch Size | 128 |
| Classifier Optimizer | Adam |
| RL Algorithm | SAC |
| **RND** | |
| Hidden Layer Sizes | $(256, 256)$ |
| Output Units | 512 |

Table 4: General hyperparameters used across all domains.

**Goal Examples:** For the classifier-based methods in our experiments (VICE and BayCRL), we provide 150 goal examples for each environment at the start of training. These are used as the pool of positive examples when training the success classifier.

**DDL Reward:** We use the version of DDL proposed in Hartikainen et al. (2019) where we provide the algorithm with the ground truth goal state $\mathbf{g}$, then run SAC with a reward function of $r(\mathbf{s}) = -d^{\pi}(\mathbf{s}, \mathbf{g})$, where $d^{\pi}$ is the learned dynamical distance function for the policy at the current iteration of training.

### A.6.1 Zigzag Maze Hyperparameters

| **BayCRL** | |
|---|---|
| Classifier Hidden Layers | [(512, 512), **(2048, 2048)**)] |
| $\lambda_{dist}$ | [**0.5**, 1] |
| $k_{query}$ | **1** |
| **VICE** | |
| $n_{\text{VICE}}$ | [1, **2**, 10] |
| Mixup $\alpha$ | [0, **1**] |
| Weight Decay $\lambda$ | [0, **5 × 10⁻³**] |
| **VICE+Count Bonus** | |
| $n_{\text{VICE}}$ | [1, **2**, 10] |
| Mixup $\alpha$ | [0, **1**] |
| Classifier reward scale | [**0.25**, 0.5, 1] |
| Weight Decay $\lambda$ | [**0**, 5 × 10⁻³] |
| **DDL** | |
| $N_d$ | [**2**, 4] |
| Training frequency (every $n$ steps) | [16, **64**] |

Table 5: Hyperparameters we tuned for the Zigzag Maze task. Bolded values are what we use for the final runs in Section 6.

### A.6.2 Spiral Maze Hyperparameters

| **BayCRL** | |
|---|---|
| Classifier Hidden Layers | [(512, 512), **(2048, 2048)**)] |
| $\lambda_{dist}$ | [**0.5**, 1] |
| $k_{query}$ | **1** |
| **VICE** | |
| $n_{\text{VICE}}$ | [1, **2**, 10] |
| Mixup $\alpha$ | [0, **1**] |
| Weight Decay $\lambda$ | [0, **5 × 10⁻³**] |
| **VICE+Count Bonus** | |
| $n_{\text{VICE}}$ | [1, **2**, 10] |
| Mixup $\alpha$ | [0, **1**] |
| Classifier reward scale | [**0.25**, 0.5, 1] |
| Weight Decay $\lambda$ | [**0**, 5 × 10⁻³] |
| **DDL** | |
| $N_d$ | [**2**, 4] |
| Training frequency (every $n$ steps) | [16, **64**] |

Table 6: Hyperparameters we tuned for the Spiral Maze task. Bolded values are what we use for the final runs in Section 6.

### A.6.3   SAWYER PUSH HYPERPARAMETERS

| **BayCRL** | |
|---|---|
| Classifier Hidden Layers | $[(512, 512), \mathbf{(2048, 2048)}]$ |
| $\lambda_{dist}$ | $[0.2, \mathbf{0.6}, 1]$ |
| $k_{query}$ | $\mathbf{1}$ |
| **VICE** | |
| $n_{\text{VICE}}$ | $[1, 2, \mathbf{10}]$ |
| Mixup $\alpha$ | $[0, \mathbf{1}]$ |
| Weight Decay $\lambda$ | $[\mathbf{0}, 5 \times 10^{-3}]$ |
| **VICE + RND** | |
| $n_{\text{VICE}}$ | $[1, 2, \mathbf{10}]$ |
| Mixup $\alpha$ | $[0, \mathbf{1}]$ |
| RND reward scale | $[\mathbf{1}, 5, 10]$ |
| **DDL** | |
| $N_d$ | $[\mathbf{4}, 10]$ |
| Training frequency (every $n$ steps) | $[\mathbf{16}, 64]$ |

Table 7: Hyperparameters we tuned for the Sawyer Push task. Bolded values are what we use for the final runs in Section 6.

### A.6.4   SAWYER PICK-AND-PLACE HYPERPARAMETERS

| **BayCRL** | |
|---|---|
| Classifier Hidden Layers | $[\mathbf{(512, 512)}, (2048, 2048)]$ |
| $\lambda_{dist}$ | $[0.2, \mathbf{0.6}, 1]$ |
| $k_{query}$ | $\mathbf{1}$ |
| **VICE** | |
| $n_{\text{VICE}}$ | $[1, \mathbf{2}, 10]$ |
| Mixup $\alpha$ | $[0, \mathbf{1}]$ |
| Weight Decay $\lambda$ | $[\mathbf{0}, 5 \times 10^{-3}]$ |
| **VICE + RND** | |
| $n_{\text{VICE}}$ | $[1, \mathbf{2}, 10]$ |
| Mixup $\alpha$ | $[0, \mathbf{1}]$ |
| RND reward scale | $[\mathbf{1}, 5, 10]$ |
| **DDL** | |
| $N_d$ | $[4, \mathbf{10}]$ |
| Training frequency (every $n$ steps) | $[\mathbf{16}, 64]$ |

Table 8: Hyperparameters we tuned for the Sawyer Pick-and-Place task. Bolded values are what we use for the final runs in Section 6.

### A.6.5 SAWYER DOOR OPENING HYPERPARAMETERS

| BayCRL | |
|---|---|
| Classifier Hidden Layers | [(**512, 512**), (2048, 2048)] |
| $\lambda_{dist}$ | $[0.05, 0.1, \mathbf{0.25}]$ |
| $k_{query}$ | $[1, \mathbf{2}]$ |
| **VICE** | |
| $n_{\text{VICE}}$ | $[1, \mathbf{5}, 10]$ |
| Mixup $\alpha$ | $[\mathbf{0}, 1]$ |
| Weight Decay $\lambda$ | $[\mathbf{0}, 5 \times 10^{-3}]$ |
| **VICE + RND** | |
| $n_{\text{VICE}}$ | $[1, \mathbf{5}, 10]$ |
| Mixup $\alpha$ | $[\mathbf{0}, 1]$ |
| RND reward scale | $[1, \mathbf{5}, 10]$ |
| **DDL** | |
| $N_d$ | $[4, \mathbf{10}]$ |
| Training frequency (every $n$ steps) | $[\mathbf{16}, 64]$ |

Table 9: Hyperparameters we tuned for the Sawyer Door Opening task. Bolded values are what we use for the final runs in Section 6.

### A.7 PROOF OF THEOREM 1 CONNECTING NML AND INVERSE COUNTS

We provide the proof of Theorem 1 here for completeness.

**Theorem A.1.** *Suppose we are estimating success probabilities $p(e = 1|s)$ in the tabular setting, where we have a separate parameter independently for each state. Let $N(s)$ denote the number of times state $s$ has been visited by the policy, and let $G(s)$ be the number of occurrences of state $s$ in the successful outcomes. Then the CNML probability $p_{CNML}(e = 1|s)$ is equal to $\frac{G(s)+1}{N(s)+G(s)+2}$. For states that are never observed to be successful, we then recover inverse counts $\frac{1}{N(s)+2}$.*

*Proof.* In the fully tabular setting, our MLE estimates for $p(O|s)$ are simply given by finding the best parameter $p_s$ for each state. The proof then proceeds by simple calculation.

For a state with $n = N(s)$ negative occurrences and $g = G(s)$ positive occurrences, the MLE estimate is simply given by $\frac{g}{n+g}$.

Now for evaluating CNML, we consider appending another instance for each class. The new parameter after appending a negative example is then $\frac{g}{n+g+1}$, which then assigns probability $\frac{n+1}{n+g+1}$ to the negative class. Similarly, after appending a positive example, the new parameter is $\frac{g+1}{n+g+1}$, so we try to assign probability $\frac{g+1}{n+g+1}$ to the positive class. Normalizing, we have

$$p_{\text{CNML}}(O = 1|s) = \frac{g + 1}{n + g + 2}. \tag{9}$$

When considering states that have only been visited on-policy, and are not included in the set of successful outcomes, then the likelihood reduces to

$$p_{\text{CNML}}(O = 1|s) = \frac{1}{n + 2}. \tag{10}$$

$\square$

