# OpenReview forum: "Reinforcement Learning with Bayesian Classifiers: Efficient Skill Learning from Outcome Examples"
_ICLR.cc/2021/Conference — Reject_

### Official Review · AnonReviewer3 · 2020-10-28

**Rating:** 5
**Confidence:** 3

**Review:**

This paper studies how to solve RL problems with a set of success states instead of a standard reward function. The central idea is to firstly train a Bayesian classifier from both the input success examples and the on-policy sampling using the conditional normalized maximum likelihood (CNML) and then use the learned classifier as a reward function to guide exploration. It is proved that in a tabular case, the success classifier trained with CNML is equivalent to a version of count-based exploration and it is claimed that with function approximation, the classifier attains non-negligible generalization. Empirically, it is claimed that this approach outperforms existing algorithms on a number of navigation and robotic manipulation domains.

The novelty of this work lies in the use of CNML to train a more regularized success classifier and further use meta-learning to implement CNML in practice.

There are several concerns I have:
1. Can the authors indicate the following information in the experiment:
- Which algorithms are provided with success examples as prior knowledge in all testing domains? The descriptions in Sec. 6.1 and 6.2 are a little confusing.
- In Figure 4, I didn’t see the lines for VICE+count-bonus.


2. The claim that BayCRL outperforms uninformed, task-agnostic exploration (VICE+count-bonus and VICE + RND ?) is not surprising since the former has prior knowledge.


3. The authors claimed that the proposed approach can achieve both effective reward-shaping and exploration. I agree with the first point by comparing it with other IRL methods. But how is the latter true? I think this needs to be further demonstrated in a different setting such as the one in the next point.


4. All the tested domains have only one success state. Thus, the example set is informational complete. I wonder about the robustness of the algorithm if the example set does not contain all success states while a reward function (which is surely super sparse) is provided. For example, if there are 10 success ground-truth states and only 5 are provided in the example set and the uncovered states are quite remote from the provided ones (but a reward function is available, i.e., if we reach these hidden success states, a positive reward will be received), then how would this degenerate the performance of the proposed approach comparing with other methods? I think this also testifies the generalization/exploration ability of the algorithm from another perspective.

I vote for weak reject since both CNML and MAML including the reformulation of the problem follow the prior works, which kind of limits the novelty of this paper as applying known algorithms to a defined problem. But I am open to adjusting the score if the rebuttal can address my concerns.

---

> ### Author Response · Authors · 2020-11-14
> **Response to Reviewer Concerns**
>
> Thank you for your insightful suggestions and comments! To address the concerns you raised, we ran several new experiments and visualizations as described in the shared response. Please find responses to questions and concerns below:
>
> > “limits the novelty of this paper as applying known algorithms to a defined problem.”
>
> The observation that a Bayesian classifier can be applied to the problem of reward inference in RL is a key novel contribution of this work. We:
> 1. Explain that CNML can serve as an appropriate bayesian method for reward inference
> 2. Instantiate an entirely novel, practical way to obtain fast CNML estimates via meta-learning (this has not been proposed > before in any algorithm — MAML is typically an algorithm for few-shot classification, and it needs to be reformulated for the problem of CNML estimation)
> 3. Theoretically and empirically show how this provides connections to exploration and reward shaping.
>
> Also, although CNML is a well-established concept, it is intractable on deep neural networks — as shown in our new table in Appendix A.3.3, our meta-CNML algorithm achieves around a 2000x speedup compared to naive CNML, and is what makes evaluation on RL problems possible. We feel that this constitutes a significant set of novel contributions.
>
> > “Experimental details:”
>
> The VICE comparisons, with and without exploration bonuses, the DDL algorithm are all provided with access to the success example. The point we make in these experiments is that BayCRL actually provides us an effective way to leverage this prior knowledge more effectively than the other techniques. We have added this description into Section 6.1 as well as the analysis in Figure 8.
>
> > “Missing comparisons to VICE + count bonus”
>
> Thank you for the pointer, we have updated Fig 5 accordingly. The VICE + count bonus lines were previously shown in an incorrect color; this is now fixed.
>
> > “The claim that BayCRL outperforms uninformed, task-agnostic exploration is unsurprising”
>
> We definitely agree that the methods mentioned perform task-agnostic exploration, but the point of our comparison here is to show that BayCRL provides a direct way to actually provide task-directed exploration. We note that VICE + an exploration bonus, which we compare to in our experiments, also has access to goal examples, but does not use it for the exploration bonus itself. We can see the advantages of our method the better empirical performance on task success in Fig 5, and the reward shaping and exploration visualizations in Fig 7 and 8.
>
> > “How does the bayCRL algorithm help with exploration?
>
> An easy way to understand the benefit of BayCRL for exploration is by noting the direct connection that BayCRL has to inverse counts, as noted in Theorem 4.1. Basically, as a state is seen more and more in the dataset and given a label of 0, the resulting likelihood that CNML assigns to it becomes lower and lower. The exact connection in the absence of function approximation is 1/(N + 2) where N is the number of times a state is visited. This connection directly reveals that the CNML likelihoods in the absence of function approximation give an inverse count bonus, which is what many exploration algorithms do [1]. To visualize this quantitatively, we also plot the number of uniquely visited states vs the number of iterations of training and see that BayCRL trained policies visit far more unique states than a standard MLE classifier (Fig 8).
>
> > “Suggestion of sparse reward + goal states (point 4)”
>
> Thank you for the suggestion! We are in the process of running this comparison and will add in these results shortly. (Update 11/19: this experiment has been added, please see the next comment below for details)
>
> [1] Unifying Count-Based Exploration and Intrinsic Motivation, Bellemare et al 2016

---

> > ### Author Response · Authors · 2020-11-19
> > **Additional Experiment and Revisions**
> >
> > We have added the results and additional analysis for your suggested experiment in point 4 (multiple sparse reward states with some not provided as goals) to the Appendix in section A.5.2. The results show that BayCRL is able to uncover “success states” that are not given to the algorithm explicitly and are very far from the provided goals, due to its built-in exploration behavior that causes it to consider multiple possible directions of movement before finding a goal. Meanwhile, prior classifier-based methods do not have this behavior and will move only towards the incomplete set of provided success states.
> >
> > However, we would like to clarify a few things about your original point:
> > - Typically, the setup for BayCRL (and classifier-based RL algorithms in general) assumes that no reward function exists, and the user provides success examples to alleviate the need for manually specifying such a reward function. However, to create a realistic setting for your proposed experiment, we can assume that there is a reward function which is not known to the human expert, and that the examples provided by the expert therefore may not cover all the regions with high reward.
> > - You mentioned using this to test the “generalization/exploration ability of the algorithm”. To clarify, the results of our new experiment demonstrate BayCRL’s exploration ability; this is what allows it to continue considering multiple unexplored pathways despite there being no goal examples in one of the directions, leading it to eventually find the sparse rewards. It does not necessarily imply an ability to generalize to very different goals which have not been provided. Intuitively, if the unknown sparse reward is very far from any provided goals in state space, we would not expect a function approximator to generalize from one goal to another. In practice, if the environment has multiple intended goals which are far from each other, the human can provide each of these as success examples as done in [1], which would cause the learned classifier to have several high-reward regions.
> >
> > [1] End-to-End Robotic Reinforcement Learning without Reward Engineering, Singh et al. 2019. https://arxiv.org/abs/1904.07854

---

> > > ### Comment · AnonReviewer3 · 2020-11-20
> > > **After reading the revision**
> > >
> > > I want to thank the authors for their efforts to address my concerns. I have been reading the revised version and I have some extra comments and questions:
> > >
> > > 1. The main algorithm is presented in section 5. Maybe a primitive version (i.e., without meta-training but only CNML) can be displayed in Section 4 to help the readers understand the content in Section 4? It's a little hard to catch the learning process.
> > >
> > > 2. In Figure 1, should the numerator of r(s) be the union with (s,1) instead of (s,0)? I thought we want r(s) = p(e=1|s).
> > >
> > > 3. When I read the paper, it keeps telling me what a good reward-shaping should be but it takes me several times to go over and collect the information about why CNML works. The information is scattered in some intuitive description about CNML on classification task in Section 3.2, an illustration in Section 4.1 second paragraph, and the relation established with count-based exploration, etc. From all the above information I gathered, the reasons are:
> > >  1) assume the states are represented by features in a meaningful metric space;
> > >  2) if the unknown states are close to either positive or negative samples then label it accordingly with higher certainty;
> > >  3) if the unknown states are not close to either negative or positive then explore with a uniform distribution,
> > >
> > > where the second point shows task-specification guidance which outperforms the task-agnostic exploration and the third point provides exploration/regularization which outperforms standard MLE. Basically, CNML is more lenient on unknown states by providing them with probability close to 0.5 instead of completely arbitrary compared with non-regularized MLE.
> > >
> > > Please correct me if there is a misunderstanding. Maybe it is better to give a more straight-to-the-point and complete explanation in one place?
> > >
> > > 4. In Section 4, the authors are using 'the' policy. Is it the policy \pi that indeed changes each iteration as in Algorithm 1 or a collection of policies that we have rolled-out in total? Since in Algorithm 1, the set S- is never cleared, I guess it collects all states that can be visited by all previous policies?
> > >
> > > 5. I saw other reviewers also asked about the seemingly reward decay question on successful states in Thm. 4.1. I think it would be better if a primitive algorithm can be presented in this section and the notations e.g., N(s) and G(s), can be defined accordingly.
> > >
> > > 6. In Section 4.3, it is stated that the classifier is akin to an exploration bonus. I personally think this is not very accurate since a bonus is something added to the true reward function but there is no original reward in this setting.
> > >
> > > I am still reading the experiment part and will update the comment later.

---

> > > > ### Author Response · Authors · 2020-11-22
> > > > **Implemented suggestions**
> > > >
> > > > Thank you very much for your detailed feedback!  We have addressed your suggestions as detailed below:
> > > >
> > > > > The main algorithm is presented in section 5. Maybe a primitive version (i.e., without meta-training but only CNML) can be displayed in Section 4 to help the readers understand
> > > >
> > > > Thanks for the suggestion — we have added pseudocode to section 4.1, describing a basic CNML-based RL algorithm without metalearning, to aid with understanding. (A diagram of classifier-based RL methods in general is also provided in Figure 1.) Please let us know if this helps make the section more clear.
> > > >
> > > > > In Figure 1, should the numerator of r(s) be the union with (s,1) instead of (s,0)? I thought we want r(s) = p(e=1|s).
> > > >
> > > > Good catch! We do indeed want r(s) = p(e=1|s) rather than p(e=0|s). This has been fixed in Figure 1.
> > > >
> > > > > When I read the paper, it keeps telling me what a good reward-shaping should be… Maybe it is better to give a more straight-to-the-point and complete explanation in one place?
> > > >
> > > > Your understanding is correct. We have added a clearer summary of the main advantages of CNML in Section 4.4, which gives an overview of all the material that was introduced over the course of Section 4.
> > > >
> > > > > In Section 4, the authors are using 'the' policy. Is it the policy \pi that indeed changes each iteration as in Algorithm 1 or a collection of policies that we have rolled-out in total? Since in Algorithm 1, the set S- is never cleared, I guess it collects all states that can be visited by all previous policies?
> > > >
> > > > In theory, our negative examples should come from the state distribution of the current policy \pi(s). However, you are right that our practical algorithm does not clear the set \mathcal{S}_- of visited states, and thus it includes states that were visited by previous policies. In fact, the practical implementation we use for \mathcal{S}_- is actually a replay buffer, from which we sample visited states as negatives for training the classifier. We based this design decision on prior classifier-based RL methods like [1].
> > > >
> > > > > I saw other reviewers also asked about the seemingly reward decay question on successful states in Thm. 4.1. I think it would be better if a primitive algorithm can be presented in this section and the notations e.g., N(s) and G(s), can be defined accordingly.
> > > >
> > > > We have added pseudocode for a basic CNML-based RL algorithm (without metalearning) in Section 4.1 as suggested. If this is still unclear after our revisions, please let us know!
> > > >
> > > > > In Section 4.3, it is stated that the classifier is akin to an exploration bonus. I personally think this is not very accurate since a bonus is something added to the true reward function but there is no original reward in this setting.
> > > >
> > > > Although there is no original reward function, we can actually think of the discrete CNML classifier rewards as a combination of a sparse reward function (which is 0 everywhere except the goal) and an exploration bonus. At non-goal states, discrete CNML would output 1/(N(s) + 2) as described in Theorem 4.1, which corresponds to a reward of 0 plus an inverse count bonus. At goal states, it would output (G(s) + 1)/(N(s) + G(s) + 2), which corresponds to the sparse reward function outputting a non-zero value for reaching the goal.
> > > >
> > > > We have clarified this point in Sections 4.2 and 4.3.
> > > >
> > > > ---
> > > >
> > > > [1] Variational Inverse Control with Events: A General Framework for Data-Driven Reward Definition, Fu et al. 2018. https://arxiv.org/abs/1805.11686

---

> > > > > ### Comment · AnonReviewer3 · 2020-11-23
> > > > > **Thanks for the revision**
> > > > >
> > > > > Thank you for revising the paper and I do think it helps me understand the algorithm better. I only have one left confusion:
> > > > >
> > > > > You mentioned that *the* policy is the current running policy instead of the collection of all previous policies. Then by Thm. 4.1, at convergence, a success state will have a reward 0.5 and a never-visit unsuccessful state under the current policy will also have a reward 0.5 (since G(s) = N(s) = 0)? Does this mean the reward encourages exploration as much as reaching the goal? Will this push the agent away from success states?
> > > > >
> > > > > If the above is true, I actually think using the collection of all previous policies, i.e., do not clear S-, makes more sense. In that way, there is a wider cover over the state space and the case of a never-visit unsuccessful state merely happens so that *at convergence*, we guarantee that the successful states can have larger rewards than unsuccessful states?

---

> > > > > > ### Author Response · Authors · 2020-11-24
> > > > > > **Regarding definition of S-**
> > > > > >
> > > > > > Good observation! You are correct that defining S- as negatives from the collection of all previous policies results in wider coverage of visited states, which is useful for our approach which must estimate uncertainty in the rewards for each state. Since this is indeed what we do in practice (representing S- as a fairly large replay buffer of previous states) as described in Algorithms 1 & 2, we are able to ensure that at convergence, the goal has reward ~0.5 which is higher than that of all other previously visited regions.

---

> > > ### Comment · AnonReviewer3 · 2020-11-21
> > > **About my initial concerns**
> > >
> > > I finished reading the experiment part of the revised version and my initial concerns are addressed. I will consider adjusting my score after communicating with other reviewers and the area chair.

---

> ### Author Response · Authors · 2020-11-19
> **Follow-up on concerns**
>
> We have added a number of additional experiments, clarifications and revisions to the paper to address the reviewers' concerns. These provide more evidence of the exploration and reward shaping benefits of BayCRL, and the added analysis helps provide a clearer picture of the behavior of the algorithm.
>
> As we have not heard from the reviewers since the beginning of the discussion period, we would like to ask whether your concerns have been addressed and whether there are any additional questions or clarifications? We have attempted to address reviewer concerns as thoroughly as possible, and we would be very happy to engage in further discussion and improvements to our work. Your feedback so far is greatly appreciated!

---

### Official Review · AnonReviewer1 · 2020-10-29
**An interesting paper, but some concerns regarding motivation and experiments**

**Rating:** 5
**Confidence:** 3

**Review:**

Summary
-------

This paper addresses a reinforcement learning problem where the reward
function is learned through a classifier that decides whether states are
successful or not based on previous examples (i.e. RL after inverse RL).
The authors show that this requires uncertainty-aware predictions, which
are difficult with neural networks. An algorithm, BayCLR, is proposed
that uses MAML to meta-learn the conditional normalized maximum
likelihood, i.e. the "maximum likelihood distribution". Connections to
the proposed algorithm and exploration methods are discussed before
using the algorithm to solve various robotics tasks.

Decision
--------

Although I liked this paper overall, I am rating it tentatively as
marginal below the acceptance threshold. The paper is very well written
and addresses a relatively clear problem (inverse RL with classifier)
with an interesting method (meta-learning CNML). I have some issues with
unclear statements in the motivation and method that should be
addressed. While the experiments provide some insight, I think the
conclusions the authors draw from them are far stronger than the results
imply.

Originality
-----------

I am not very familiar with CNML, but this paper seems very original. In
particular, the application of meta-learning to conditional normalized
seems novel, as well as its application to inverse RL.

Quality and Clarity
-------------------

The paper is well-written, most statements are clear and easy to follow.

Strengths
---------

-   The approach of meta-learning CNML is interesting, and if anything
    deserves further analysis outside of inverse RL / RL.
-   Although I have some issues with the motivation (see below), I think
    the authors do a good job of explaining their rationale for using
    CNML. In particular, quantifying neural network uncertainty through
    posterior analysis is quite difficult.
-   The experiments seem comprehensive. However, I am not very familiar
    with the environment suite used. Due to the lack of work in this
    area, there are not many possible baselines, and so the VICE
    baseline seems like the best choice. The baseline is further kept
    fair by adding exploration heuristics. I especially like Section 6.2
    that analyzes BayCRL on the zigzag maze task, which I assume is a
    tabular environment. In addition, there is a simple ablation but I
    would prefer more work on this.

Weaknesses
----------

-   In Section 4, some motivating statements are unclear to me (see
    Detailed Comments).

-   A single data-point being added to the dataset may not change the
    distribution much, and this crucial point is only addressed (in an
    ad-hoc way) in the appendix. It would be good to see an ablation
    study on this, or perhaps a plot of the average difference between
    different query points. Perhaps Figure 6 may indirectly explain
    this, but it should be explicitly addressed.

-   In Section 6, many statements about BayCLR seem stronger than the
    results imply. Many confidence intervals overlap, and the results
    themselves are hard to parse with so many lines in each plot.
    Perhaps some of these baselines which are not competitive can be
    excluded, or perhaps different linestyles should be used.

Detailed comments
-----------------

-   Early mentions of exploration (i.e. in the abstract) seem out of
    place. While you elaborate on the connection to exploration methods,
    it does not seem like the main point of this paper is to address
    exploration. It seems to me that you are tackling a novel RL problem
    where the task is specified through goal states. Specifically, you
    learn classifier in place of a reward function, and this is
    exploited to shape an otherwise sparse reward.

-   Section 4.1, "To create effective shaping, we need to impose a prior
    on our classifier so that it provides a more informative reward when
    evaluated at rarely vis- ited states that lie on the path to
    successful outcomes."

    Why is a prior strictly necessary for reward shaping? Unless you
    mean prior in a very general sense, not a Bayesian prior, I don't
    see why a prior is strictly necessary.

-   Section 4.1, "\[CNML\].. is essentially imposing a uniform prior
    over the space of possible outcomes". This is not obvious to me, and
    perhaps further explanation is needed.

-   Section 4.2, Theorem 4.1: Perhaps I misunderstand but if $G(s) > 0$,
    shouldn't $p(e = 1 | s) = 1$? Further, why is that when the agent
    visit a successful state, i.e. $N(s)$ increases, $p(e = 1 | s)$
    decreases after each visit.

-   Section 5.1, "This algorithm, which we call meta-NML, allows us to
    obtain normalized likelihood estimates without having to retrain
    maximum likelihood to convergence at every single query point, since
    the model can now solve maximum likelihood problems of this form
    very quickly. "

    Is it correct to say that meta-NML does not need to retrain to
    convergence at every query point? The second part of the sentence
    elaborates that metatraining allows you to solve the problem very
    quickly, but it seems that you still need to solve it at every query
    point.

-   Section 6.2, Figure 4: I don't think its fair to say that BayCLR
    performs substantially better. The confidence intervals overlap in
    all but Spiral Maze and Sawyer 3d Pick-and-Place. Other subtleties
    are not addressed, such as why RND/count-bonus actually hurt VICE in
    sawyer 2d push. Other statements such as "significantly more
    efficiently" need explanation as well.

-   Section 6.3, Figure 5: for reproducibility, you should include
    exactly how many gradient steps are used in the model without
    meta-learning.

-   Section 6.4, Figure 6: I'm unfamiliar with the environment being
    used, so some additional details explaining what z means would be
    helpful.

    "Furthermore, meta-NML is able to reasonably approximate the
    idealized NML rewards with just one gradient step…"

    How is this shown in Figure 6? I don't see anything showing the
    idealized NML rewards.

Minor Comments
--------------

-   Section 4, Line 3: missing space: "ples.For example,"

Post Rebuttal
--------------

After reading the comments by the other reviewers, I have decided to keep my score at a 5. The authors reply, and the updated manuscript, helped my understanding of the paper. I was considering raising my score, however, the reviewers were nearly unanimous in their confusion regarding the framework or application of CNML. For future iterations of the paper, I suggest that the authors describe CNML, event-based control and their connection more explicitly. If the main contribution is using CNML as the classifier in event-based control, then it would also help to conduct experiments on meta-learning CNML in a supervised learning setting to further elucidate its effectiveness in the reinforcement learning application. I think your paper is very interesting, and I hope that the authors are able to use this feedback to improve their paper.

---

> ### Author Response · Authors · 2020-11-14
> **Response to Reviewer Concerns**
>
> Thank you for your thoughtful review and suggestions! Please find specific responses below
>
> > “Ablation study on impact of single datapoint added in CNML”
>
> As noted by the literature on CNML, while adding a single datapoint does not have a huge effect in parts of the state space that have a significant density of points, it causes a significant change in the predicted likelihood for low density regions of the space. To visualize this , we present both a qualitative and quantitative assessment. In Fig 3, we show the estimated classifier likelihoods given by the meta-NML model in state space before and after taking one gradient step. The large difference, especially in regions with low density of states shows the impact of the single datapoint. This is also summed up quantitatively in Fig 12, where we can see that the average difference between the MLE and the NML distribution.
>
> > “statements about BayCRL seem stronger than the results imply.”
>
> While the results in Fig 5 of the original submission do show some improvement of the baselines, they are not quite representative of the actual difference in behaviors of the resulting learned policies. To make this more clear, we plot the success rate of learned behaviors in Fig 5 instead of the previous metric of distance to the goal. This metric more clearly shows the reality of the resulting behavior - whereas the baseline algorithms often get stuck in spurious local optima (not actually achieving success, but potentially having a misleadingly small distance to the goal), BayCRL is consistently more successful.
>
> > “Why is a prior strictly necessary for reward shaping?”
>
> Without imposing an appropriate prior, the maximum likelihood classifier can converge to  an arbitrarily sharp decision boundary, with no reward shaping. With a naively chosen prior such as L2 regularization, the shaping can be arbitrarily poor and prone to local optima, as indicated by the poor performance of VICE on our experimental evaluations. With a Bayesian classifier trained with NML, the reward adaptively encourages exploration (see Theorem 4.1, Fig 3, and Fig 8) and provides reward shaping towards the goal (see Fig 7).
>
> > “"[CNML].. is essentially imposing a uniform prior over the space of possible outcomes". Further explanation is needed.”
>
> An intuitive way to understand this is by noting that for a query point x, that has no replicas in the dataset the augmented datasets D U (x, 0) and D U (x, 1) can both fit the pseudo-label perfectly, thereby giving a uniform likelihood over outcomes at 0.5. As more instances of x are encountered, the likelihood is more skewed but the additional pseudo-labelled point essentially smooths the predictions towards uniform. Theorem 4.1 formalizes this intuition in the case of tabular representations.
>
> > “if G(s)>0, shouldn't p(e=1|s)=1?”
>
> Good observation! In fact, the way that they algorithm is structured, at convergence p(e=1|s) for successful outcomes would be 0.5 (balanced large numbers of positives and negatives), and every other state will have p(e=1|s) will be 0. So this still encourages the agent to go towards the goal, and provides the agent with the correct gradient.
>
> > “Does meta-NML need to retrain to convergence at every query point?”
>
> We are not sure if this is asking about meta-training or taking gradient steps to evaluate a point according to CNML, so we will address both:
>
> Meta-training: while in principle meta-NML would retrain with every new datapoint that is obtained, in practice we retrain meta-NML once every k epochs. We warm-start the meta-learner parameters from the previous iteration of meta-learning, so every instance of meta-training only requires a few steps of fine-tuning.  We have added these details to Appendix A.2.
>
> Taking gradient steps to evaluate a query point: because our model is meta-trained to adapt quickly to arbitrary inputs, at test time we can take just a single gradient step rather than training to convergence and still obtain a good approximation of the NML outputs. This is the key advantage of using our method. We have clarified this in the paper by adding visualizations in Fig 3 and Appendix A.3.1.
>
>
> > Details for reproducibility and environment details:
>
> We added a clarification to our ablation experiment — we take 1 gradient step using the same method as meta-NML, just without any meta-training beforehand. We have also added more environment details in Appendix A.4.
>
> > Visualization of idealized NML rewards
>
> We have updated the visualization in Fig 3, where we can now see the similarity between the idealized NML and meta-NML rewards.
>
> > “analyzes BayCRL on the zigzag maze task, which I assume is a tabular environment”
>
> To clarify, the maze task is not a tabular environment; the state space is continuous. (described in Appendix A.4.) However, in our visualizations of the maze, we evaluate the reward at discrete intervals of the state space to show the reward contours.

---

> > ### Comment · AnonReviewer1 · 2020-11-20
> > **Two brief comments**
> >
> > Thank you for your detailed reply. This indeed addresses many of my
> > initial concerns. Below are some general comments with respect to your
> > reply and the shared response.
> >
> > -   Single datapoint impact: My original concern was the detail in
> >     Appendix A.2.1 on importance weighting used for the query point. It
> >     is not immediately obvious how this relates to low density states
> >     (as someone not familiar with CNML literature), and the connection
> >     to Figure 3 and the figures in the Appendix A.3.1.
> >
> > -   p(e = 1 | s ) " at convergence p(e=1|s) for successful outcomes
> >     would be 0.5 (balanced large numbers of positives and negatives),
> >     and every other state will have p(e=1|s) will be 0. So this still
> >     encourages the agent to go towards the goal, and provides the agent
> >     with the correct gradient."
> >
> >     How does this encourage the agent to go to the goal if the rewards
> >     are 0 everywhere except at the states that are successful
> >     outcomes? Also, what is the connection to this and gradients?

---

> > > ### Author Response · Authors · 2020-11-22
> > > **Response to Additional Questions**
> > >
> > > We appreciate your additional feedback! To respond to your questions:
> > >
> > > > Single datapoint impact: My original concern was the detail in Appendix A.2.1 on importance weighting used for the query point. It is not immediately obvious how this relates to low density states (as someone not familiar with CNML literature), and the connection to Figure 3 and the figures in the Appendix A.3.1.
> > >
> > > Our use of importance weighting as described in Appendix A.2.1 is intended to lower the variance while training with SGD. When taking gradient steps at test time for CNML, we care about the adapted model’s outputs on the single query point x_q, yet x_q will not be present in most of the batches during stochastic optimization. While in principle training to convergence would still allow our model to perform well on the query point, due to imperfections in optimization, the query point would likely be “drowned out” by the original dataset. By instead including the query point in every batch and downweighting appropriately, the objective remains unbiased but lower variance, since the query point is in every gradient update.
> > >
> > > We have added a more detailed description of this in Appendix A.2.1 as well as a plot comparing the performance of gradient descent with and without importance sampling. The version with importance weighting learns significantly faster and does not have as much variance in losses across batches. Please let us know if this helps!
> > >
> > > As for the connection to the points you mentioned:
> > > - *Low density states*: there is no explicit connection to low density states, since importance weighting is meant to help with adapting to any query point in general. However, low density query points do occur less frequently in the training distribution, so reducing variance of the MLE training process by including them in every batch ensures that adaptation can happen consistently for those states as well.
> > > - *Figure 3 and Appendix A.3.1*: both of these use importance weighting, i.e. the query point is downweighted and included in every batch at test time. Had we not done this, the progression in A.3.1 would have been less stable, and some of the steps may have brought us closer to the MLE solution (a sparse reward) rather than the true CNML solution if those batches did not happen to include the query point.
> > >
> > > > How does this encourage the agent to go to the goal if the rewards are 0 everywhere except at the states that are successful outcomes? Also, what is the connection to this and gradients?
> > >
> > > The rewards are only 0 everywhere except the goals *at convergence*, i.e. if the agent has found the goal and the policy no longer changes much between iterations. Importantly, we don’t train the classifier to convergence immediately and use it to assign all future rewards; instead, we jointly optimize the policy (using a standard RL algorithm) and the classifier (by training on visited states and goal examples), so the rewards are continually changing based on the data collected by the agent. A general progression would be something like this:
> > > - The rewards are initially very smooth, and increase in the direction of the goal examples, providing guidance on which direction to go
> > > - As the agent visits more states, the rewards for those visited states drop. However, as long as the goal has not yet been reached, there will still be shaping towards the goal
> > > - After the agent has found the goal and is able to consistently reach it, most other states will have their rewards brought down to 0 and the goal will have reward close to 0.5. At this point, we have converged to a sparse reward function, but this is okay because the learned policy is able to reach the goal as desired and should no longer be altered.
> > >
> > > We note that this idea is not originated by us and is common in the inverse RL literature. Previously, [1] showed a connection between GAN-style training and maximum entropy IRL, both of which have discriminators (or reward classifiers) which output 0.5 at convergence on the desired distribution of states. This was then generalized in [2] to the case where only goal examples are provided. Regarding the connection to gradients, we apologize for the confusing wording. The more precise way to say this would be: since we train our classifier on visited states and goal examples, under reasonable assumptions about generalization and an appropriately regularized classifier, the output success probability (and therefore the reward) would be a smoothly increasing function as we get closer to the goal.
> > >
> > > --
> > >
> > > [1] A Connection between Generative Adversarial Networks, Inverse Reinforcement Learning, and Energy-Based Models, Finn et al. 2016. https://arxiv.org/abs/1611.03852
> > >
> > > [2] Variational Inverse Control with Events: A General Framework for Data-Driven Reward Definition, Fu et al. 2018. https://arxiv.org/abs/1805.11686

---

> > > > ### Comment · AnonReviewer1 · 2020-11-25
> > > > **Thank you for these clarifications**
> > > >
> > > > Thank you for these clarifications. I will re-evaluate the paper in the light of these details, the revisions and your discussions between the other reviewers.

---

> ### Author Response · Authors · 2020-11-19
> **Follow-up on concerns**
>
> We have added a number of additional experiments, clarifications and revisions to the paper to address the reviewers' concerns. These provide more evidence of the exploration and reward shaping benefits of BayCRL, and the added analysis helps provide a clearer picture of the behavior of the algorithm.
>
> As we have not heard from the reviewers since the beginning of the discussion period, we would like to ask whether your concerns have been addressed and whether there are any additional questions or clarifications? We have attempted to address reviewer concerns as thoroughly as possible, and we would be very happy to engage in further discussion and improvements to our work. Your feedback so far is greatly appreciated!

---

### Official Review · AnonReviewer4 · 2020-10-29
**This paper aims to better address a known RL problem that learns reward from a set of successful events**

**Rating:** 4
**Confidence:** 4

**Review:**

This manuscript aims to solve reinforcement learning problems where the reward is unknown but a set of successful states are available. Iteratively, it trains a classifier using provided successful states as positive and on-policy samples as negative and use its predictions as the reward function to learn RL policy.

It may be worth clearly explaining the connection and improvement on the work of Fu et al.(2018b, Variational inverse control with events: A general framework for data-driven reward definition) in introduction since both mainly solve the same problems. Fu’s paper first introduced the event framework to generalize inverse RL that also solves the sparse RL problems by iteratively training a classifier to predict the probability on successful states.

Compared to previous work, it seems a major difference in this paper is to change the event classification model, replacing the neural network classifier with CNML classifier. It could enhance the contribution if  good theoretical analysis can be provided. E.g. why the manuscript concludes that CNML performs better than standard neural network clasisifers in terms of uncertainty aware on reward estimation, and how CNML model connects with better exploration and goal-oriented reward shaping.

Other comments are written below.

Theorem 4.1 is defined, but it seems not used in the following sections. It would be helpful if the following can explain whether it is used  to develop the algorithm or explain empirical findings.

It may need more evidence in Section 4.3 to support the conclusion that the CNML is doing better at reward shaping. It is not easy to make a conclusion based on some specific example in Figure 1. Similarly Section 4.1 also try to show CNML gives reward that can improve exploration and becomes goal oriented using the same example.

Section 5.1 describes how to use meta learning to approximately solve CNML problem so it can reduce computation complexity. It is an interesting idea, but it may need more analysis. Some questions are listed below.

- The distribution of tasks usually has different samples, while the CNML problem has similar data sets for its tasks as each task only adds one sample to the original dataset. Does this difference have influence on learning the model parameter in meta learning?

- It would be interesting to analyze the computation complexity as it is the main reason for applying meta learning in solving CNML.

In experiments, It may be interesting to test the CNML estimation with full NN convergence on entire augmented dataset every time, to show the trade-off between accuracy and computation complexity when compared with meta learning.

Based on the number of epochs, it seems BayCRL converges slower in some problems, such as zigzag, spiral, sawyer. It may be interesting to give some insights, as good reward may speed the learning process. Moreover, it may be more accurate to consider the time complexity in each epoch when validating the time complexity.

On page 7, it may need explanation on how to find that BayCRL outputperforms in terms of sample comcplexity as shown in Figure 4.

Some minor comments:
- On page 5, Appendix Appendix A.5 -> Appendix A.5
- It seems Appendix A.2 duplicates Section 5.1, particularly the Eq. 6 and the equations for p_meta-NML and \theta_y.
- Is there any reason for line 7 in Algorithm 1 that skips meta learning.

---

> ### Author Response · Authors · 2020-11-14
> **Response to Reviewer Concerns**
>
> Thank you for your comments and feedback! We have added new visualizations and experiments as described in the shared response. Please find specific responses below:
>
> > “explaining the connection and improvement on the work of Fu et al.”
>
> We have added a clarification to Section 6.1. The key difference is indeed the fact that the classifier is trained via CNML, rather than a standard classifier trained with MLE and L2 regularization. As seen in results comparing VICE (Fu et al) and BayCRL (ours) in Fig 5, this makes a significant difference in learning progress and improves reward shaping and exploration.
>
> > “Connection of Theorem 4.1 to develop the algorithm or explain empirical findings.”
>
> Theorem 4.1 connects the Bayesian classifier to exploration, since it shows that the classifier rewards are equivalent to inverse counts as exploration bonuses in the absence of meaningful generalization. This provides intuition for why BayCRL is able to solve challenging exploration tasks such as the mazes. To illustrate this connection, we visualize the rewards in Fig 13 in Appendix A.3.2. (In practice, BayCRL does not simply output inverse counts; the goal examples and generalization additionally allows for better shaping towards the goal, as Fig 13 shows.)
>
> We have also added a new quantitative metric in terms of the number of states visited by BayCRL versus a standard MLE classifier, available in Fig 8. We see that BayCRL avoids getting stuck in local optima and more quickly visits states in the direction of the goal, indicating the exploration benefits suggested by Theorem 4.1.
>
> > “evidence for CNML is doing better at reward shaping.”
>
> Evidence for the reward shaping of CNML is shown qualitatively in Fig 7, as well as empirically through our comparisons in Fig 5 to a prior classifier reward method (VICE, Fu et al 2018) with exploration bonuses. The fact that BayCRL performs better indicates that it is not simply playing the role of an inverse count bonus, but is also providing better shaping towards the goal.
>
> > “evidence to show CNML gives reward that can improve exploration”
>
> We have added a quantitative plot (Fig 8) that shows the number of unique states visited using BayCRL as opposed to a standard MLE classifier. We can see that even when the actual reward is not observed, BayCRL encourages the agent to explore and visit additional states. The example in Section 4.1 is simply for schematic illustration; empirical evidence is provided in Fig 8 and the improvements in Fig 5 over standard VICE. This is also reinforced theoretically by Theorem 4.1.
>
> > “each task only adds one sample to the original dataset. Does this difference have influence on learning the model parameter in meta learning?”:
>
> Yes! First, we note that the one sample point can be thought of as the actual "task", since the goal is to adapt to that point quickly. The rest of the points are included simply so that the meta-training update corresponds to a step of SGD on the overall dataset, allowing us to take additional steps to converge to CNML if desired.
>
> In practice, the single datapoint does have a big impact on the learned parameters. To illustrate this we visualize pre-update and post-update predictions in Fig 3, which makes it clear that the outputs are significantly different. We also show how the average absolute difference between pre and post-update predictions increases with additional gradient steps in Fig 12 in the Appendix.
>
> > computation complexity of meta-CNML”
>
> We have added a table of runtimes for standard feedforward inference, meta-NML, and naive CNML in section A.3.3 of the Appendix. These results demonstrate the significant speedup of meta-CNML over naive CNML, which is key to making our method practical.
>
> > “test the CNML estimation with full NN convergence on entire augmented dataset”
>
> Unfortunately, running an entire RL experiment with CNML trained to convergence would be impractical, as each epoch would take several hours as noted in section A.3.3. However, we hope the table of runtimes along with the visual and quantitative analysis of meta-NML performance with varying numbers of gradient steps in A.3.1 make the tradeoffs clear.
>
> > “consider the time complexity in each epoch when validating the time complexity.”
>
> Appendix A.3.3 shows the time complexity per epoch for our classifier schemes. We note that while the runtimes of meta-NML are still slower than feedforward inference (i.e. VICE), in our results, VICE requires far more epochs to converge (resulting in more time taken), or does not solve the task at all.
>
> > “explanation on how BayCRL outperforms in terms of sample complexity”
>
> We have added a better visualization of performance via success rate in Fig 5, which paints a clearer picture than the plots in the submission.  In general, BayCRL is the only method that solves all of the tasks consistently, and even if other methods converge earlier, they converge to a suboptimal solution or spurious local optima.

---

> ### Author Response · Authors · 2020-11-19
> **Follow-up on concerns**
>
> We have added a number of additional experiments, clarifications and revisions to the paper to address the reviewers' concerns. These provide more evidence of the exploration and reward shaping benefits of BayCRL, and the added analysis helps provide a clearer picture of the behavior of the algorithm.
>
> As we have not heard from the reviewers since the beginning of the discussion period, we would like to ask whether your concerns have been addressed and whether there are any additional questions or clarifications? We have attempted to address reviewer concerns as thoroughly as possible, and we would be very happy to engage in further discussion and improvements to our work. Your feedback so far is greatly appreciated!

---

> > ### Comment · AnonReviewer4 · 2020-11-23
> > **Thanks for the response**
> >
> > Thanks for the detailed response. Though the empirical results look promising on maze tasks, the paper could benefit from convincing analysis that shows how the CNML advances the exploration and goal-oriented reward shaping for general RL problems. Therefore, I may retain my original score.

---

> > > ### Author Response · Authors · 2020-11-24
> > > **Clarifying evidence for exploration and reward shaping**
> > >
> > > Thanks for your feedback! Can you please clarify why the empirical results are only promising on the maze tasks and not on the Sawyer robot environments? The intent of the three Sawyer environments was to evaluate our method on tasks that represent a wider variety of general RL problems on which classifier-based algorithms would be useful — all of them involve fairly difficult exploration and require nontrivial reward shaping to solve the task with standard methods. Were there other types of general RL problems you were hoping to see?
> > >
> > > We would also like to emphasize that we have included what we believe is an overwhelming amount of empirical evidence for how our algorithm "advances the exploration and goal-oriented reward shaping for general RL problems":
> > > - Theorem 4.1 establishes a connection between CNML and count-based exploration in the discrete case, without any generalization across states.
> > > - Figure 5 in Section 6.2 shows that BayCRL outperforms existing algorithms on navigation and robotic manipulation, providing evidence for improvements in both reward shaping and exploration.
> > > - (*Added in response to reviews*) Figure 7 and the “BayCRL and Reward Shaping” analysis in Section 6.4 visualizes the rewards given by CNML on the challenging 3D pick-and-place task as compared to a standard MLE classifier.
> > > - (*Added in response to reviews*) Figure 8 and the “BayCRL and Exploration” analysis in Section 6.4 looks at the state coverage of BayCRL vs. standard classifier-based methods both visually and quantitatively.
> > > - (*Added in response to reviews*) Appendix A.3.2 compares the reward shaping given by various classifier methods on the maze task qualitatively.
> > > - (*Added in response to reviews*) The ablation in Appendix A.5.1 shows how even in a discrete environment BayCRL reduces to count-based exploration.

---

### Official Review · AnonReviewer2 · 2020-11-05
**some interesting ideas but require deeper analysis**

**Rating:** 5
**Confidence:** 4

**Review:**

This paper considers the problem of learning a policy for an MDP with unspecified reward, given user-provided goal states. To this end, a reward model and a policy are jointly learned: the reward model is the conditional normalized maximum likelihood (CNML) learned from a training set consisting of the example goal states as positive examples, and the policy trajectories as negative examples; the policy is trained to optimize the MDP using the learned reward. Meta-learning is applied to reduce the cost of learning the CNML models.

Pros
+ The idea of using CNML to obtain a smoother reward as compared to a single model (together with the efficient meta-learning approximation) is interesting.
+ The algorithm is compared with several baselines and seem to perform well. Ablation study suggests that goal examples and meta-learning are important for the proposed approach.

Cons
- The paper is unnecessarily hard to read. A high-level description of the approach early in the paper will be helpful. This is also related to the comments below: it is not clear why the algorithm should work.
- A claimed contribution of the paper is to "produce more tractable RL problems and solve more challenging classes of tasks". The limited feedback provided by the example goal states perhaps make the RL problem more tractable, but how is it possible to solve more challenging classes of problems when less information is available?
- To learn a useful reward model from the example goal states, the CNML approach alone seems insufficient, and it seems necessary to require a good reward to be a smooth function of feature vectors. For example, if we work in a grid world with random rewards, does the approach still work?
- Both the paragraph before Sec 3.2 and Alg 1 mention that the set of negative examples keeps growing. This implies that the reward model will become more and more sparser (values closer to zero), even for the goal states? How is such a reward model still useful?
- Another question about the reward model is that when the policy becomes better, it is more likely to reach the goal states, thus the goal states are more likely to be labeled as both positive and negative. Thus the reward model is more likely to assign lower reward to goal states when more training is done?
- Fig. 1 seems to be overstating the problem with MLE. What are the features used and what is the classifier model? If the feature is the real-valued position, and a regularized logistic regression model is used, then MLE will not produce such a sparse reward as in (b)?
- The experiments section should provide more details about the experimental setup: the choice of candidate classifier models, explanation of the baselines (e.g. Sparse Reward seems not mentioned in the text at all), detailed description of the performance evaluation metric. Is Manhattan distance to goal a sensible performance metric for maze navigation?

Minor comments
- "OpenAI et al.": wrong citation format
- Define L in Eq. (2)

Post-rebuttal
After reading the rebuttal and other reviewers' comments, my score remained the same. The rebuttal helped to clarify some issues, but it is still not clear to me why the algorithm should work. I agree with other reviewers that a more careful revision of the paper, and a further analysis on the algorithm will be beneficial.

---

> ### Author Response · Authors · 2020-11-14
> **Response to Reviewer Concerns**
>
> Thank you for your comments and suggestions! We have performed a number of new experiments and visualizations to address your concerns, which have been added to the paper as described in the shared response. Please find detailed responses to questions below:
>
> > “A high-level description of the approach early in the paper will be helpful“:
>
> We have added a high level overview section (Section 4.4), and made the overview figure which explains the entire pipeline appear earlier and more prominently. Please let us know if this helps with clarity of the exposition in Section 4.
>
> > “how is it possible to solve more challenging classes of problems when less information is available?”:
>
> Excellent question! The intuition is that the general class of RL problems can have arbitrary (sparse) reward, and in the worst case would need complete state coverage. Our approach assumes that we know what successful outcomes look like. When this information is provided, the algorithm can extract a signal that guides it towards success, whereas in the most general RL problem, the reward may be completely uninformative until the agent randomly chances upon a successful outcome. For example, a standard RL agent in a 2D grid may need to exhaustively explore the state space before observing a reward. However, successful outcomes given at the start of training can use these examples to know the direction to go. BayCRL can exploit this extra information, whereas standard RL cannot. We can see this type of behavior in our analysis of BayCRL in Section 6.4.
>
> > “seems necessary to require a good reward to be a smooth function of feature vectors”:
>
> For the reward shaping that results from the classifier to be meaningful, the feature vectors need to be smooth and structured such that a CNML classifier would be able to appropriately extract shaped reward. In practice, this is the case with many RL environments such as those we consider in Section 6. However, even if the features are not structured and completely lack identity, then the CNML classifier would still reduce to an exploration-centric reward bonus, as indicated by Theorem 4.1, ensuring reasonable worst-case performance. We are in the process of running an empirical validation of this and will post results shortly. (Update 11/17: this experiment has been added, please see the next comment for details)
>
> > “the reward model will become more and more sparse , even for the goal states?”
>
> We will clarify this in the text. The dataset for training is balanced to have an equal number of positives and negatives, sampled from the entire set of positives and negatives seen thus far. If the policy learns to perfectly reach the goal states, then the rewards will be pushed down to zero and the goal states will converge to 0.5 since there will be a roughly equal number of positives and negatives at the goal. However, as long as the policy is imperfect, the rewards will not converge to this sharp reward, ensuring that the agent continues towards the goal. As noted briefly in Section 3.1, prior work has shown that such a scheme corresponds to optimizing a generalized inverse reinforcement learning objective, and is a principled way to do reward inference.
>
> > “Fig. 1 seems to be overstating the problem with MLE. Logistic regression with regularization would not provide sparse reward”
>
> Good observation! We will correct the scope of claims and note that Fig 1 is a conceptual diagram meant to portray the issues with an unregularized MLE model. In the case with standard regularization such as L2 regularization, we get a reward that is dependent on the particular form of the regularizer, and can often lead to spurious local optima on more complicated examples. On the other hand, an adaptive CNML based regularizer allows for much better shaping as well as inherent exploration to avoid getting stuck in these local optima. To illustrate this, we have added a few figures to our paper:
> - Figure 8 shows the visitations on the spiral maze task for BayCRL as compared to VICE (Fu et al), which does use an L2 regularized classifier, and demonstrates how VICE gets stuck on the other side of the wall. We note that this is also reflected in the resulting performance in our experimental plots in Fig 5.
> - Appendix A.3.2 which compares additional classifier training schemes, including both unregularized and regularized MLE, on a concrete 2D maze dataset.
>
> > “The experiments section should provide more details”
>
> We have added a detailed description of the baselines and evaluation metrics in Appendix A.4 of the updated paper.  We use standard multi-layer perceptrons for the classifier, trained with Adam using standard hyperparameters. Manhattan distance in this setup is referring to the shortest distance along *valid* paths to the goal, respecting the walls. This is reflective of good performance, and it will correctly identify if an agent is in a local optimum.

---

> > ### Author Response · Authors · 2020-11-18
> > **Additional Experiment and Revisions**
> >
> > We have added an experiment in the Additional Ablations section (Appendix A.5.1) to address your question about the performance of BayCRL in a grid world environment without smooth feature vectors. In this new variant of the Zigzag Maze task, states are first discretized to a 16×16 grid, then "shuffled" so that the xy representation of a state does not correspond to its true coordinates and the states are not correlated dynamically. We see that BayCRL still solves the task, whereas a standard classifier method fails. This provides empirical justification for Theorem 4.1 which states that in the absence of any generalization across states, CNML reduces to a count-based exploration bonus, ensuring that the agent continues to explore until it finds the goal.
> >
> > However, when this performance is compared with the performance using semantically meaningful input features, as described in our main results (comparable plot in Appendix A.4.2, Fig 14), we see that the performance with shuffled states is not as consistent. This suggests that the performance of BayCRL cannot simply be attributed to exploration but also to the reward shaping induced by generalization with the Bayesian classifier. We note that this is not a limitation in practice, because in most practical continuous domains (including the ones we consider in our paper), we can expect some degree of smoothness in the features of the environment rather than the completely random gridworld we have constructed in this ablation.
> >
> > Additionally, we have added a clarification in Section 6.1 of the paper, indicating that the comparisons with sparse reward and L2 reward refer to running the base Soft Actor-Critic algorithm with reward being either sparse or L2. This comparison is meant to show that even with the algorithm kept the same, the form of the reward is particularly important for facilitating learning.

---

> ### Author Response · Authors · 2020-11-19
> **Follow-up on concerns**
>
> We have added a number of additional experiments, clarifications and revisions to the paper to address the reviewers' concerns. These provide more evidence of the exploration and reward shaping benefits of BayCRL, and the added analysis helps provide a clearer picture of the behavior of the algorithm.
>
> As we have not heard from the reviewers since the beginning of the discussion period, we would like to ask whether your concerns have been addressed and whether there are any additional questions or clarifications? We have attempted to address reviewer concerns as thoroughly as possible, and we would be very happy to engage in further discussion and improvements to our work. Your feedback so far is greatly appreciated!

---

### Author Response · Authors · 2020-11-14
**Shared Response to Reviewers**

We thank the reviewers for their thoughtful comments and suggestions. To address the concerns brought up by reviewers, we conducted a number of additional experiments, and edited the paper for clarity and exposition.  We describe these below

Changes in text:

We added a more clear method overview in Section 1, added significantly more details about the experiments and baselines in Section 6.1 and Appendix A.4, made clarifications about the expected behavior of the algorithm in Section 5.2, and added in clarifications about the computational complexity of meta-NML in Table 1 and Appendix A.3.3. We have also toned down the language in Section 6.2 describing the performance of BayCRL compared to prior algorithms, so that the improvements can be interpreted more objectively from the plots instead.

New experiments and Visualizations:

- **Computational efficiency:** To demonstrate the substantial efficiency improvement of our meta-learned CNML approach compared to standard CNML, we computed the runtime per input point and per epoch of RL for the two methods. These runtimes are available in the tables in Appendix A.3.3. Meta-CNML achieves a speedup of 1600 to 2300 times the runtime of CNML from scratch.

- **Single datapoint impact:** We also visualize the impact of a single added datapoint in training, which is quite substantial as seen in Fig 3 as well as Fig 11 & 12 in Appendix A.3.1, both qualitatively and quantitatively.

- **Quantitative visualization of exploration:** We added visualizations of trajectories and state coverage on the spiral maze task for BayCRL vs. a normal classifier training scheme (VICE (Fu et al 2018)) in Fig 8. We also introduce a better quantitative evaluation of exploration, showing the fraction of states visited with and without using a Bayesian classifier. We see that using the Bayesian classifier increases the number of states visited by approximately 30%, and is also directed towards the goal rather than exploring all states naively.

- **Additional visualizations of meta-NML:** We have provided additional comparisons of the classifiers resulting from meta-NML as compared to ideal NML, regularized MLE, and unregularized MLE in Fig 13 in Appendix A.3.2. These new figures show that meta-NML better guides the agent towards the goal by rewarding underexplored states. We also added a visualization of the rewards assigned by our meta-CNML model before and after taking a gradient step at evaluation time, showing the impact of adaptation. We also added a plot of average difference between MLE and CNML classifier goal probabilities as we take more gradient steps for CNML in Fig 12.

- **More clear plotting of experiments:** To make the experimental comparisons to other RL algorithms easier to understand, we plot the success rate in Figure 4 instead of a distance metric over time. These plots are less cluttered and more representative of overall performance, since many of the baseline methods were actually getting stuck in local optima and the distance metric may not make this clear. The original distance plots are available in Appendix A.4.2, along with more detailed explanations of the success thresholds and distance metrics used.

- **Learning in a discrete, randomized environment:** following the suggestion of Reviewer 2, we have added a new experiment in Appendix A.5.1 demonstrating BayCRL's ability to learn in environments where the states are not smooth or correlated dynamically, due to the worst-case performance guaranteed by its exploration effects.

- **Finding "hidden" rewards not indicated by success examples:** as suggested by Reviewer 3, we have added a new experiment in Appendix A.5.2 where the environment has multiple potential goals, some of which are not provided to the algorithm as success examples. We see that BayCRL is able to consider multiple directions of movement and find the most convenient sparse rewards to reach (despite not knowing about those states initially) due to its built-in exploration ability, whereas prior classifier-based methods fail to do so.

- **Details on importance weighting for CNML adaptation:** We have provided additional justification in Appendix A.2.1 for the importance weighting scheme used during CNML test-time adaptation, as well as a plot showing its concrete improvements to learning stability compared to standard minibatch gradient descent.

---

### Decision · Program_Chairs · 2021-01-07
**Final Decision**

**Decision:**

Reject

**Comment:**

Summary:
This paper introduces a method to try to learn in environments where a person specifies successful outcomes  but there is no environmental reward signal.

I'd personally be interested in knowing where people were able to easily provide such successful outcomes instead of, for instance, providing demonstrations or reward feedback. Similarly, I'd be interested in how other methods of providing human prior knowledge compared.

Discussion:
Reviewers agreed the paper was interesting, but none of the 4 thought the paper should be accepted.

Recommendation:
While I do not think this paper should be accepted in its current form, I hope the authors will find the comments and constructive criticism useful.